# Human milk oligosaccharide mediates mutualism between *Escherichia coli* and *Bifidobacterium bifidum*

David Seki [1,2] ✉, Shaul Pollak [2], Magdalena Kujawska [1,3], Raymond Kiu [3,4], Antia Acuna-Gonzalez[4], Lucy I. Crouch [3], Cassie R. Bakshani[3], Peter T. Chivers [5,6], Monique Mommers[7], Nils van Best [8], John Penders [8] & Lindsay J. Hall [1,3,4,9] ✉

Infant gut microbiota development involves frequent colonization by *Enterobacteriaceae*, particularly *Escherichia coli*, yet their ecological role in healthy infants is unclear. Here, we analyse longitudinal stool samples from healthy, term-born, breastfed infants (n = 41) and related mothers (n = 30) using shotgun metagenomics and novel computational approaches. Strain-resolved profiling indicates that *Bifidobacterium* species are frequently shared within families, whereas *E. coli* derive from external sources, but often persist within individuals. Despite differing ecological strategies, these genera co-exist and share evolutionary adaptations related to lactose acquisition in the infant gut. In vitro, we demonstrate that interactions between *E. coli* and *Bifidobacterium bifidum* are mutualistic in co-culture, where *E. coli* supplies cysteine to its auxotrophic partner, facilitating cooperative degradation of 2′-fucosyllactose, the predominant human milk oligosaccharide. In turn, the liberated monosaccharides sustain *E. coli* growth, highlighting a cooperative cross-feeding interaction that may contribute to regulating *E. coli* abundance within the infant host.

Infant health is tied to the postnatal assembly of the gut microbiota. Despite major research interest[1–4], our understanding of the principles underlying microbial community assembly is still lacking. This is largely due to difficulties in identifying true microbe-microbe and host-microbe interactions in natural populations, whose dynamics arise from the interplay between deterministic factors such as diet[5] and geographical location[6] with countless other stochastic components.

Human breastmilk represents the primary source of nutrition during early life. Its most abundant carbohydrate is lactose, which provides nutrition for the infant, although it can also be metabolized by a wide range of gut-associated bacterial taxa[7]. To sustain an intestinal population of commensal microbes, lactose is additionally incorporated within greater oligosaccharide structures, collectively known as human milk oligosaccharides (HMOs). HMOs are not absorbed in the proximal gut and thus remain available to microbial

[1]Intestinal Microbiome, School of Life Sciences, ZIEL - Institute for Food & Health, Technical University of Munich, Freising, Germany. [2]Department of Microbiology and Ecosystem Science, Division of Microbial Ecology, Centre for Microbiology and Environmental Systems Science, University of Vienna, Vienna, Austria. [3]Department of Microbes, Infection and Microbiomes, School of Infection, Inflammation and Immunology, College of Medicine and Health, University of Birmingham, Birmingham, UK. [4]Food, Microbiome and Health, Quadram Institute Bioscience, Norwich, UK. [5]Department of Biosciences, University of Durham, Durham, UK. [6]Department of Chemistry, University of Durham, Durham, UK. [7]Department of Epidemiology, Caphri School of Public Health and Primary Care, Maastricht University, Maastricht, The Netherlands. [8]Department of Medical Microbiology, Infectious Diseases and Infection Prevention, NUTRIM School of Nutrition and Translational Research in Metabolism, Maastricht University Medical Center+, Maastricht, The Netherlands. [9]Norwich Medical School, University of East Anglia, Norwich, UK. ✉e-mail: david.seki@univie.ac.at; l.hall.3@bham.ac.uk

strains with the enzymatic capacity to degrade them[8]. Together, these features impose two key ecological constraints on infant gut colonization: (i) the ability to capture and degrade HMOs, and (ii) competition for accessible mono- and disaccharides.

*Bifidobacterium* frequently dominates the gut microbiota of breastfed infants[9]. Their presence has been associated with several beneficial functions, including induction of immunological tolerance[10], establishment of colonisation resistance against pathogens[11], improved vaccine response during the first year of life[12], and maintenance of gut-barrier integrity[11]. This early dominance is supported by their rich set of glycoside hydrolases enabling efficient HMO-degradation[13] and direct maternal transfer[14]. At the same time, *Escherichia coli* is commonly detected in breastfed infants, typically at low relative abundances but with high prevalence across individuals[10,15–22]. *E. coli* are generally considered pathobionts[23], as upon overgrowth, they can cause infectious diarrhoea, which remains a significant cause of neonatal sepsis and mortality[24], particularly in low- and middle-income countries[25]. *E. coli* lacks the capacity to degrade HMOs[26], but its generalist metabolic strategy and rapid turnover of mono- and disaccharides are thought to underlie colonization of the intestinal ecosystem[27–29]. In breastmilk, soluble lactose is not completely absorbed in the small intestine[30] and may therefore contribute to *E. coli* persistence. Additionally, *E. coli* may access simple sugars by cross-feeding interactions with extracellular degraders of HMOs, two mechanisms that are not mutually exclusive. Milk oligosaccharide cross-feeding has been reported between *Bacteroides* and *E. coli*[17], but it remains unclear if *Bifidobacterium* could similarly provide accessible sugars to *E. coli*.

To investigate ecological factors associated with infant gut community composition, we performed metagenomic sequencing of stool samples from a Dutch cohort of healthy neonates ($n = 41$). Notably, these infants represent a selected subset of the larger LucKi Birth cohort[31], chosen for their rare status of exclusive breastfeeding post-delivery before gradually transitioning to a more complex diet during the first year. This well-defined dietary trajectory provides a unique model system to investigate organizing principles of infant microbiome community dynamics. To explore it in depth, we developed a new computational pipeline - MAJIC (Mean across Jaccard index checkerboards), and profiled microdiversity of infant and maternal gut microbiota, revealing ecological aspects for the transmission of strains among individual hosts. Furthermore, without relying on annotation of genes, we identified ecological links between *Bifidobacterium* and *Escherichia* concerning their strategies to acquire lactose, supported by their consistent co-occurrence in neonates. Lastly, our experimental findings demonstrate that in co-culture, *E. coli* actively supplies cysteine to auxotrophic *B. bifidum*, facilitating the cooperative breakdown of 2'-fucosyllactose (2'FL), one of the predominant HMOs. This in vitro interaction highlights a cross-feeding mechanism that may contribute to the regulation of *E. coli* persistence and abundance within the infant host.

## Results

### *E. coli* and *Bifidobacterium* co-occur consistently during early-life

Although the order of colonizing bacteria is well described[4,32,33], the mechanisms driving assembly of the early-life gut microbiome remain poorly understood. Here, we employed shotgun metagenomics to sequence 78 longitudinal stool samples from 41 healthy neonates. These included an early time-point ($n = 18$) at 2 months post delivery, during which all infants were exclusively breastfed. Additional samples were collected at 6 months post delivery ($n = 25$), a time-point that commonly marks the onset of weaning[34], and a late time-point at 11 months post delivery ($n = 35$), by which complex foods were introduced. In comparison to the broader LucKi Birth cohort[31], where 74.1% of infants are no longer exclusively breastfed by 6 months, our cohort

represents a rare subset. Additionally, to evaluate potentially relevant components of vertical transfer between mother and infant we sequenced 30 stool samples of respective mothers at 2 weeks post delivery (Fig. 1A and Supplementary Data 1).

Overall, alpha diversity was similar at 2 and 6 months post-delivery (Shannon, *t*.test, *p*.adj = 0.061) and increased significantly thereafter (Shannon, *t*.test, *p*.adj <0.001, respectively; Figs. 1B and S1A). For between-sample comparisons, species present in <10% of samples at a minimum relative abundance of 0.1% within each age group were excluded to improve statistical robustness and reduce noise. Principal component analysis (PCA) indicated that infant gut microbiotas were most dissimilar to each other at 2 months post delivery, gradually becoming more similar to same-aged infants and to maternal microbiotas over time (Fig. 1C). This trend was more pronounced between infants and their own mothers compared to unrelated mothers (Fig. S1B). Microbiome composition varied significantly with age (PERMANOVA, *p*.adj <0.05), except between 2 and 6 months post delivery (PERMANOVA, *p*.adj = 0.731), highlighting the selective pressure exerted by a breastmilk-based diet. *Bifidobacterium breve* was on average the most abundant species in infants, followed by *Bifidobacterium bifidum*, and *Bifidobacterium longum* subspecies *longum*. As long as breastmilk was supplied, *E. coli* occurred at low relative abundance, but consistently (74% and 96% prevalence at 2 and 6 months post delivery), alongside initially dominant *Bifidobacterium*. In maternal samples, *Bifidobacterium adolescentis* was the most abundant bifidobacterial species on average (Figs. 1D and S1C, and Supplementary Data 2). To assess the distribution of taxa across infants, we calculated a "cross-sample Shannon diversity" index (CSI) for each species, quantifying the evenness of the species' abundance across all samples. Prevalent *Bifidobacterium* species, including *B. bifidum* (CSI = 3.8), *B. longum* subsp. *longum* (CSI = 3.8), *B. breve* (CSI = 3.5), and *B. adolescentis* (CSI = 3.4), ranked within the top 5% of this distribution (>95th percentile), indicating broad and even prevalence across the cohort. In contrast, other *Bifidobacterium* species (e.g., *B. animalis, B. catenulatum*) exhibited markedly lower CSI, reflecting restricted occurrence and uneven abundance patterns.

In the majority of cases, *E. coli* co-occurred with either a single dominant *Bifidobacterium* species or multiple highly abundant *Bifidobacterium* species during the initial 6 months post delivery (Figs. 1E and S1D, E). However, at 2 months postpartum, the relative abundance of *E. coli* was notably higher when only one single *Bifidobacterium* species was highly abundant (relative abundance > 20%) compared to when multiple *Bifidobacterium* species were highly abundant. But this difference did not remain statistically significant after correction for multiple testing (*t*-test, *p* = 0.043, *p*.adj=0.14; Fig. S1F). In rare cases where *E. coli* was present without a highly abundant *Bifidobacterium*, the following alternative species were so instead: at 2 months post delivery, *Clostridium perfringens* (45%-abundance) and *Bifidobacterium pseudocatenulatum* (53%), once each. At 6 months post delivery, *Ruminococcus gnavus* (42%-abundance), *Bacteroides fragilis* (31%-abundance), *B. pseudocatenulatum* (29%), and *Bacteroides clarus* (42%-abundance) in four separate cases (Fig. S1G). Notably, all of these species are evidentially capable of degrading HMOs[35–37], while *E. coli* lack this ability[38].

Together, our findings show that *E. coli* and *Bifidobacterium* coexist in the gut of most breastfed infants. However, *E. coli* is not restricted to communities dominated by extracellular HMO degraders (e.g., *B. bifidum*), but also appear alongside species that primarily import HMOs for intracellular degradation (e.g., *B. longum* subsp. *longum*, *B. breve*)[39] (Fig. S1D). This implies that residual lactose may be sufficient to sustain their co-existence.

### Mean across Jaccard index checkerboards (MAJIC)

Before focusing specifically on interactions between *Bifidobacterium* and *E. coli*, we first aimed to capture the broader landscape of putative

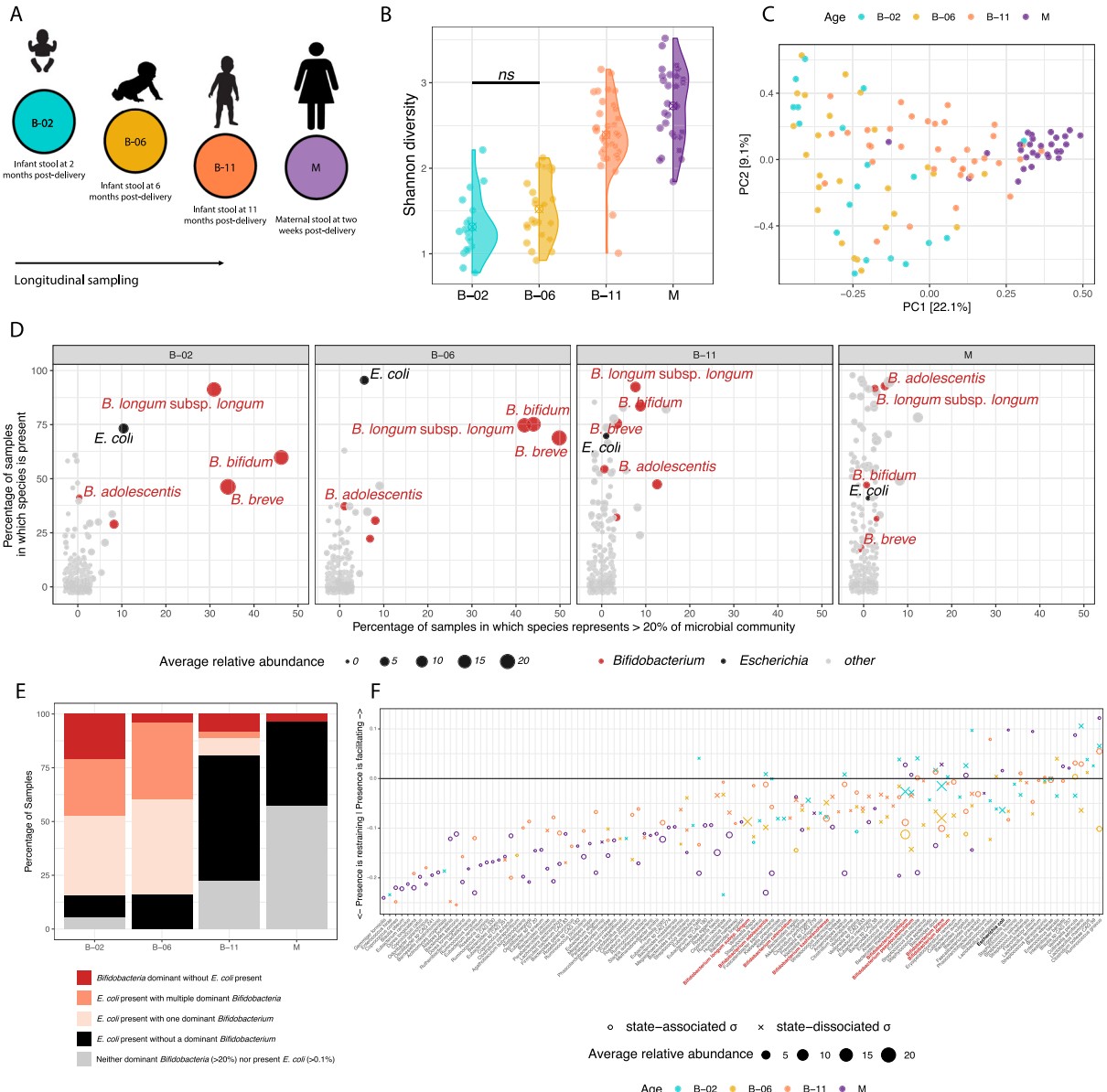

**Fig. 1 | Characteristics of infant gut microbiota during the initial year post-delivery at species-level resolution. A** Overview of study-design. Samples were obtained from babies at two-, six-, and eleven months post-delivery (B-02, B-06, B-11), and one maternal sample was obtained two weeks post-delivery (M). **B** Microbial richness measured by Shannon diversity (Two-sided unpaired *t*-test with FDR correction). **C** Principal component analysis (PCA) of relative species counts over Hellinger distance. **D** Scatterplot highlighting prevalence (percentage of samples in which species are present) and high abundance (percentage of samples in which species represent >20% of the microbial community) for *Bifidobacterium* and *Escherichia coli* in maternal and infant samples. **E** Bar plot displaying the distribution of microbial colonization patterns across sample categories. Bars represent the percentage of samples (y-axis) classified according to different combinations of abundances between *Bifidobacterium* (including *B. adolescentis*, *B. breve*, *B. bifidum*, and *B. longum* subsp. *longum*) and *E. coli*. **F** Differences in mean Jaccard dissimilarity $\Delta\mu_i = \mu_i^+ - \mu_i^-$, wherein negative values indicate that microbiota are more restrictive when focal species is missing [$\sigma_i^- (\mu_i^-)$], and conversely more facilitating when $\Delta\mu_i > 0$, therefore focal species being present [$\sigma_i^+ (\mu_i^+)$]. It was identified via comparisons to shuffled data whether focal species ($\sigma_i$) is state-associated (X) or state-dissociated (O), and colour indicates age-groupings. On the x-axis, *Bifidobacterium* are highlighted in red and *E. coli* in black. ⊗ in violin plots indicate group median.

interactions within the infant gut microbiota on the species level. To this end, we developed a computational pipeline - MAJIC - to systematically assess the extent to which focal microbes may influence gut ecology. Our approach involves comparing microbial communities in which focal species (i) is present ($\sigma_i^+$) to those where it is absent ($\sigma_i^-$). To ensure robust statistical testing, we generate 10,000 random $\sigma_i^+$ and $\sigma_i^-$ datasets by shuffling the presence/absence labels of focal species (i) each time. This randomization scheme accounts for differences in sample sizes and abundance distributions of individual species. For each focal species, we calculate the mean Jaccard dissimilarity of samples in $\sigma_i^+$ ($\mu_i^+$), and in $\sigma_i^-$ ($\mu_i^-$), as well as the difference in mean Jaccard dissimilarity ($\Delta\mu_i = \mu_i^+ - \mu_i^-$), comparing these values to the random distributions generated from shuffled data. A negative $\Delta\mu_i$ ($<0$) indicates that microbiotas are more constrained in the presence of the focal species (since $\mu_i^+ < \mu_i^-$), whereas a positive $\Delta\mu_i$ ($>0$) indicates greater community variability when the focal species is present. Furthermore, we define a species as 'state-associated' when the distribution of its $\mu_i^+$ values, computed from observed data, differs significantly from that obtained by shuffled inputs (ks.test, $p < 0.05$). Conversely, "state-dissociated" species show no significant difference ($p \geq 0.05$), suggesting no strong association with specific community states (Fig. S2A).

At 2 months post delivery, infant microbiotas demonstrated a high degree of compositional flexibility, which was gradually lost over time. By 11 months, and in maternal microbiotas, communities became more compositionally constrained, with an increased prevalence of "state-associated" focal species, compared to earlier time points (Fig. S2B). We identified *R. gnavus* as the most facilitating species in maternal microbiota ($\Delta\mu = 0.12$), and *Bacteroides stercoris* as the most restrictive species in infant microbiota 11 months post delivery ($\Delta\mu = -0.25$), suggesting critical roles in niche occupation for these putative keystone species. Interestingly, highly abundant bifidobacteria (*B. breve*, *B. bifidum*, and *B. longum* subsp. *longum* and *B. adolescentis*) were classified as "state-dissociated" at 2 months post delivery, suggesting no strong association with specific community states. This suggests that early post-delivery, despite their recognised importance for infant development and gut microbiome assembly, highly abundant *Bifidobacterium* species largely do not function as putative keystone species in the classical ecological sense[40]. Instead, their loss is likely compensated by functional equivalent species, maintaining overall composition and functional stability despite an individual species' absence at 2 months post delivery. However, after 6 months post delivery *B. bifidum* became "state-associated", and so did *B. adolescentis* and *B. breve* 11 months post delivery (Fig. 1F).

Our analysis furthermore extends beyond presence/absence patterns to identify pairs of state-associated microbes that quantitatively influence each other's abundance within the community. For each focal species (i) as defined above, we identified any species (j) whose abundance patterns significantly differed between matrices containing species (i) ($\sigma_i^+$) and those without it (i) ($\sigma_i^-$) using a Wilcoxon rank-sum test. At 2 months post delivery, most significant associations occurred among "state-dissociated" species, suggesting that these interactions were likely stochastic. In contrast, significant associations between "state-associated" species were rare but consistently negative across all time points (Fig. S2C), indicating that deterministic, competitive dynamics are limited to a small number of taxa.

Lastly, we filter our data to identify any non-stochastic instances where a "state-associated" focal species ($\sigma_i$) exerts a significant effect on any type of species (j) with involvements of dominant *Bifidobacterium* or *E coli*. Thereby, we found no significant interactions at 2 and 6 months post delivery. At 11 months post delivery, *B. adolescentis* was significantly more abundant in the presence of *Gemella haemolysans*. In maternal samples, *B. breve* was negatively affected by the presence of *Lactobacillus fermentum* and *Enterococcus gallinarum*. Importantly, across all timepoints we found no significant evidence of

antagonism between highly-abundant *Bifidobacterium* and *E. coli* using our pipeline. However, given that they consistently co-occur in the infant gut, we conclude that their relationship must be ecologically neutral and that their co-occurrence is likely driven by shared environmental preferences and permissible niche overlap, rather than direct interactions. In contrast, *E. coli* exhibited significantly reduced abundance in the presence of *Klebsiella pneumoniae* at 11 months post delivery, pointing to a potential antagonistic relationship between these two *Enterobacteriaceae* (Fig. S2D).

## Strain-resolved transmission routes in infant gut microbiota

To better understand the ecological strategies and transmission dynamics of key early-life gut microbes, we next investigated strain-level variation and gene-level diversity within highly-abundant *Bifidobacterium* species and *E. coli*. Therefore, we assembled metagenome-assembled genomes (MAGs) from shotgun sequencing data and overall acquired 458 unique dereplicated high-quality genomes (Supplementary Data 3). On average, these MAGs accounted for $75 \pm 14\%$ of metagenomic reads in infants at 2 months, $78 \pm 5\%$ at 6 months, and $75 \pm 7\%$ at 11 months post delivery, while covering $68 \pm 7\%$ of maternal reads. To verify that sequencing depth was sufficient for robust genome detection, we examined the relationship between genome coverage (average read depth) and breadth (fraction of the genome covered by reads)[41]. The resulting curve plateaued near ~0.85 breadth, indicating that additional sequencing would yield little new genomic information and that most genomes, including low-abundance taxa, were captured (Fig. S3A). Using recommended thresholds which minimize false positive detection from spurious read mappings (breadth > 0.5; coverage > 5)[41], *E. coli* detection closely matched read-based profiles (Fig. S3B), confirming prevalence in infants at 2 and 6 months. Subsequently, we mapped all reads to our set of dereplicated genomes, and inStrain[41] was used to profile the nucleotide diversity ($\pi$), assessing both within (intra-$\pi$) and across (inter-$\pi$) individual variation.

Intra-$\pi$ for all MAGs did not correlate to the coverage of MAGs within samples (Spearman, $p$.adj > 0.05, respectively, Fig S3C), indicating that strain-level genetic variation in the infant gut is shaped by factors other than relative abundances of respective MAGs. Overall, intra-$\pi$ remained stable between infants at 2 and 6 months post delivery, but increased significantly thereafter ($t$.test, $p$.adj < 0.05 for every other pairwise comparison), and *E. coli* MAGs most notably displayed elevated intra-$\pi$, suggesting the presence of multiple strains co-existing within individuals, whereas MAGs of highly-abundant *Bifidobacterium* (*B. bifidum*, *B. longum* subsp. *longum*, *B. breve*, and *B. adolescentis*) rarely showed such patterns, indicating reduced strain-level diversity (Fig. 2A).

To investigate strain-level transmission patterns, we applied a threshold of pop-ANI 99.5% to distinguish between identical strains and strain-variants, and quantified strain-sharing events between individuals. Identical *B. adolescentis* strains were shared among 22% of maternal microbiota, while *B. longum* subsp. *longum* strains were shared among 31% of infants at 11 months post delivery (Fig. S3D). Furthermore, pop-ANI values were significantly higher between parenting mothers and their infants, as compared to random mother-infant pairs at two and 11 months post delivery ($t$-test, $p$.adj <0.0001 and 0.008, respectively; Fig. S3E). Notably, *B. longum* subsp. *longum* strains were frequently shared between infants and their mothers, whereas no such co-occurrence was observed for *E. coli* strains in mother-infant pairs (Fig. S3F). Next, we assessed strain persistence within infants across time for dominant *Bifidobacterium* and *E. coli*. Between 2 and 6 months post delivery, identical *B. bifidum* and *B. longum* subsp. *longum* strains persisted within individuals in 12.5% of cases, while no such persistence was observed for *B. breve*. From 6 to 11 months, identical *E. coli* strains persisted in 10% of cases, and from two to eleven, as well as from 6 to 11 months, *B. longum subsp. longum*

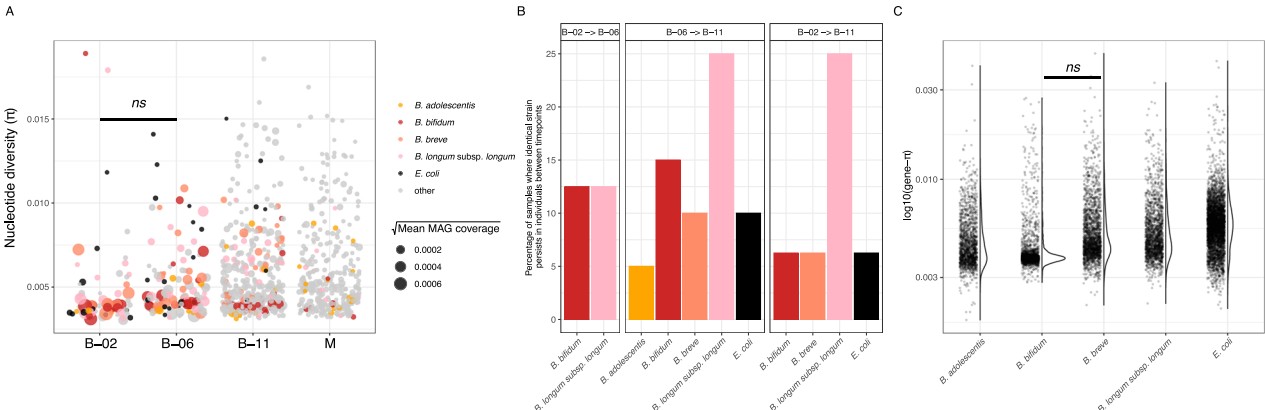

**Fig. 2 | Microdiversity and strain-sharing among infant and maternal gut microbiota. A** Dot-plot visualizing the distribution of overall nucleotide diversity (π) according to age-groups. Each dot represents a metagenome-assembled-genome (MAG) dereplicated at 95% average nucleotide identity (ANI), and size of dots represent mean coverage MAGs. MAGs of *B. bifidum, B. breve, B. longum* subsp. *longum, B. adolescentis* and *E. coli* are coloured in red, salmon, pink, orange, and black. Statistical significance was assessed using a two-sided Welch's *t*-test with FDR correction. **B** Count of strain-persistence events where identical strains of *B. bifidum, B. longum* subsp. *longum, B. breve, B. adolescentis,* and *E. coli* re-occurred between adjacent timepoints within the same individual. **C** Violin / Dot-plot visualizing the distribution of overall gene nucleotide diversity (gene-π) in *B. bifidum, B. breve, B. longum* subsp. *longum, B. adolescentis,* and *E. coli.* Each dot represents a gene in respective MAGs. Statistical significance was assessed using a two-sided Welch's *t*-test with FDR correction.

strains persisted in 25% of infants (Fig. 2B). Overall, these findings reveal distinct transmission and persistence patterns among key infant gut microorganisms. *B. adolescentis* was frequently shared among maternal microbiota, while *B. longum* subsp. *longum* strains were commonly transmitted between infants and parenting mothers. In contrast, *B. breve* and *B. bifidum* were less frequently transmitted between mothers and their respective infants but were prevalent in the infant gut. *E. coli* displayed high microdiversity overall, and although not shared between mothers and infants, strains often persisted within infants across the first year post-delivery, implying other origins of colonization.

We hypothesized that the observed differences in host-associated transmission patterns would be reflected in the genetic nucleotide diversity (gene-π) within *Bifidobacterium* and *E. coli*. Specifically, given that *E. coli* exhibited limited mother-to-infant transmission as well as higher pop-ANI values, we expected weaker host-specific selective pressures, leading to higher nucleotide diversity across a broader range of genes. Conversely, for *Bifidobacterium*, where frequent strain transmission and persistence between hosts occur, we anticipated stronger purifying selection on genes essential for host adaptation. To identify these genes, we applied EggNOG-mapper v6 for functional annotation[42], and profiled gene-π in *B. longum* subsp. *longum, B. breve, B. bifidum, B. adolescentis*, and *E.* coli. As expected, *E. coli* displayed the highest gene-π on average. Additionally, gene-π did not significantly differ between *B. breve, B. longum* subsp. *longum* (*t*.test, *p*.adj > 0.05), suggesting that despite differences in transmission dynamics, their ecological roles and adaptive strategies are similar. However, *B. bifidum* exhibited a particularly narrow gene-π distribution that significantly differed from the remaining species (*t*.test, *p*.adj < 0.0001, Fig. 2C), indicating stronger genetic constraints on its adaption. Next, we categorized genes with gene-π values above the 99th percentile as "loose" and those below the 1st percentile as "restricted." These labels are operational definitions intended to distinguish genes with unusually high or low within-species nucleotide diversity, rather than to imply predefined functional roles. Among the loose genes, we identified, for example, a starch-binding outer membrane protein from the SusD/RagB family in *B. bifidum*, which may contribute to HMO transport. Restricted genes, aside from several housekeeping functions, included a putative chitinase in *B. bifidum* and a multiple-sugar transporter in *B. breve*, both potentially involved in HMO acquisition (Supplementary Data 4).

In summary, our findings reveal distinct ecological strategies among key infant gut microbes. *B. longum* subsp. *longum* was frequently transmitted and persisted between infants and mothers, while *B. bifidum* exhibited strong genetic constraints, presumably reflecting host-specific adaptation. In contrast, *E. coli* showed high microdiversity, limited transmission from parenting mothers, but high persistence of singular strains in individual infants.

## GH-driven co-evolution underpins microbial adaptation to HMOs

We next used our MAGs to investigate functional traits underpinning microbial adaptation to the infant gut environment. Lactose and HMOs, as the primary carbon sources for early colonizing microbes, represent key ecological drivers of primary succession in the infant gut[43]. To overcome the lack of traditional gene annotation and gain a more holistic understanding of microbial traits essential for the acquisition of carbohydrates in the infant gut, we employed a recently developed annotation-agnostic approach to infer microbial trophic strategies from genomic data[44]. We hypothesized that glycoside hydrolases (GHs) are necessary for the degradation of HMOs and lactose, that they are costly to produce, and that they benefit not only the GH producers but also all potential consumers of the resulting degradation products. As such, GHs function as model public goods, where their utility extends beyond enzyme synthesis to encompass a suite of associated traits, largely unknown but likely including chemotaxis, biofilm formation, and membrane transport, that enhance the efficient capture and utilization of GH-derived breakdown products. Public goods like GHs are subject to rapid evolutionary shifts between parasitism and cooperation due to trade-offs between production costs and competitive advantages. Furthermore, their evolutionary trajectories are shaped by high rates of gene deletion and horizontal transfer. By leveraging these dynamics, we can trace patterns of co-evolution between the GHs (public goods) and battery-traits across microbial species with distinct ecological strategies, providing deeper insights into how microbial communities adapt to HMO-rich environments[44].

Because of their central role in infant gut microbiota, we sought to identify protein families that have co-evolved with GH2 family enzymes across our set of 458 unique dereplicated genomes. In total, we identified 2803 such protein families ($\varepsilon_{GH2}$), with an average of and $302 \pm 284$ per genome ($\varepsilon^i_{GH2}$). To further investigate the degree of

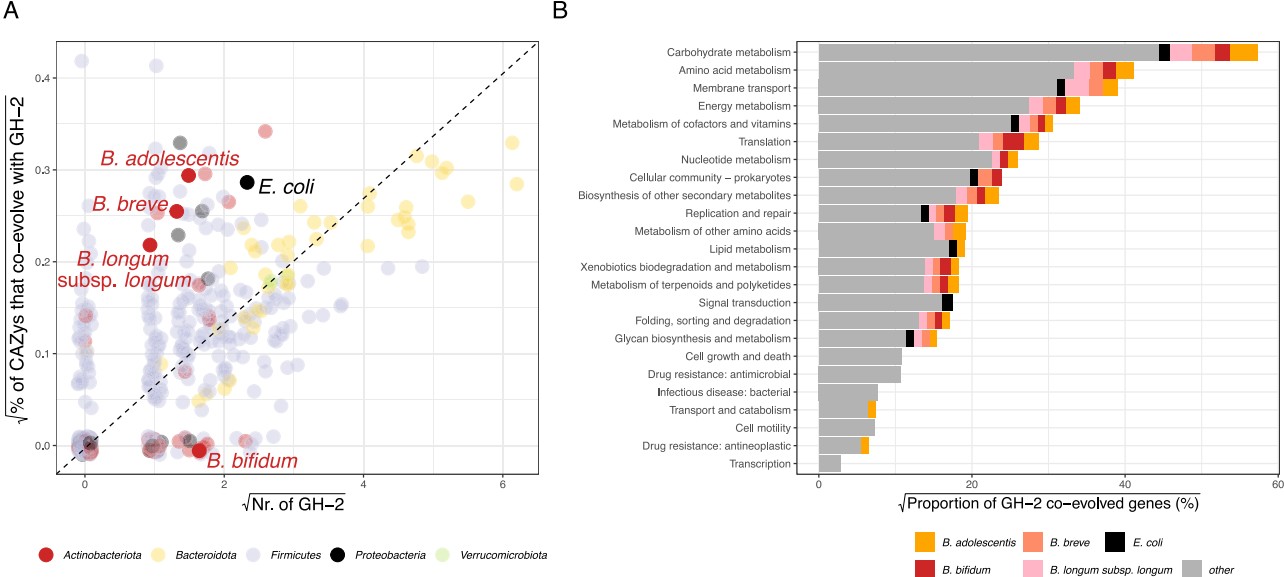

**Fig. 3 | Coevolution of microbiota with GH2 domain-containing enzymes.**
**A** Scattering of observed number of GH2 domain-containing enzymes against fraction of coevolved genes that are other carbohydrate active enzymes as well. Each dot represents a MAG, coloured by corresponding phylum. The most abundant bifidobacteria across infant and maternal samples (*B. bifidum, B. breve, B. longum* subsp. *longum, B. adolescentis)* as well as *E. coli*, are highlighted. **B** Bar-charts summarizing taxonomic origin of genes that co-evolved with GH-2 glycoside hydrolases. Bars represent the proportion (%) of GH-2 co-evolved genes assigned to different bacterial taxa, including *B. adolescentis* (orange), *B. bifidum* (red), *B. breve* (salmon), *B. longum* subsp. *longum* (pink), and *E. coli* (black). Only coevolved genes that were mapped to kegg-orthology (ko) groups via EggNOG mapper v6 are plotted (unknown were excluded).

polymer degradation specialization in different microbial lineages and the associated strategies, we compared the number of GH2 enzymes encoded in a genome ($O_{GH2}$) to two derived quantities based on the co-evolutionary patterns of GH2-related genes encoded within single genomes. The first, $E_{GH2}$, was estimated using an elastic-net linear regression model fitted across all genomes. This value represents the expected number of GH2 enzymes a genome 'should' encode based on its genomic background of $\varepsilon^i_{GH2}$. If true coevolution were occurring, $E_{GH2}$ and $O_{GH2}$ would be strongly correlated, as recently demonstrated for chitinases and their coevolved genes[44]. Indeed, we observed a strong correlation between $E_{GH2}$ and $O_{GH2}$, supporting the validity of our approach and the genuine co-evolution of $\varepsilon_{GH2}$ across diverse microbial phyla (Spearman, $R = 0.83$, $p < 0.00001$). However, our data also suggest that the specialized organisms for GH2 breakdown product utilization (high $E_{GH2}$ and high $O_{GH2}$) primarily include species from the genera *Bacteroides* and *Parabacteroides*, rather than typically dominant members of the infant gut microbiome (Supplementary Data 3).

The final parameter we examined, $f^i_{CAZy}$, represents the fraction of genes coevolving with GH2 that are themselves CAZymes. If the selection of coevolved genes were random, the correlation between $O_{GH2}$ and $f_{CAZy}$ would be expected to follow a diagonal line of identity. However, our analysis revealed a notable deviation, with *Bifidobacterium* forming a distinct cluster, alongside species from *E. coli*. We speculate that this clustering suggests a non-random coevolutionary dynamic, likely driven by shared selective pressures that require an expanded repertoire of CAZymes for efficient colonization of the infant gut. These findings highlight putative functional associations between these species. Notably, *B. bifidum* deviated from this pattern, as its GH2-associated genes were exclusively non-CAZymes, suggesting an alternative metabolic adaptation (Fig. 3A).

To further explore the functional significance of these coevolved genes, we annotated them using EggNOG-mapper v6[42]. Notably, while all genera displayed GH2 coevolving genes that could be attributed to amino acid metabolism, *E. coli* uniquely lacked such associations (Figs. 3B and S4A, B). This suggests that while *E. coli* mirrors the dynamics of *Bifidobacterium*, which are specialized in HMO-degradation, it adopts the ecological role of a versatile prototroph, and may additionally be less reliant on simultaneous amino acid degradation to access GH2 breakdown products, despite possessing the functional capacity to do so[45].

## *B. bifidum* and *E. coli* cooperate over the degradation of 2'FL

*E. coli* is unable to degrade HMOs[26], but its ability to rapidly scavenge simple mono- and disaccharides, residual lactose in particular, likely explains its prevalence in breastfed infants. Another source of simple sugars is formed by extracellular degradation of HMOs, and notably, *E. coli* co-occurs with extracellularly degrading *B. bifidum* (Fig. S1D). However, it is unknown whether external degradation of HMOs by *B. bifidum* could lead to growth of *E. coli*.

To test this hypothesis, we focused on the metabolism of 2'-O-fucosyl-lactose (2'FL), the most abundant breastmilk HMO[46]. First, we isolated *B. bifidum* from infant stool and assessed its growth on 0.1% 2'FL (w/v) as the primary carbon source in modified M9-medium (m-M9)[47]. *B. bifidum* was able to grow at 37 °C under anoxic conditions (Fig. 4A), but not in m-M9 lacking 2'FL (Fig. S5A). This confirmed that the additives to m-M9 were insufficient to support growth without an external carbon source. *B. bifidum* also failed to grow in m-M9 + 0.1% 2'FL without casamino acids or cysteine, consistent with the well-established cysteine requirement for *Bifidobacterium*[48], and indicating additional metabolic dependencies in this isolate (Fig. S5B). Additionally, we isolated highly-abundant HMO-importers from infant stool (*B. longum* subsp. *longum and B. breve*), but neither grew on m-M9 + 2'FL (Fig. 4A). We then analyzed whether *E. coli* K-12 could directly metabolize 2'FL or its individual monosaccharides - fucose, galactose, and glucose as carbon sources. While *E. coli* K-12 could not metabolize 2'FL, it exhibited slight growth in m-M9 only, in the absence of 2'FL, suggesting either a preference for amino acid metabolism in the absence of polysaccharides or possibly inhibitory effects of 2'FL on growth (Fig. S5C). However, *E. coli* K-12 efficiently grew when provided with either one of the three monomers, including fucose (Fig. S5D). Next, we tested whether *E. coli* K-12 could grow in a spent medium from

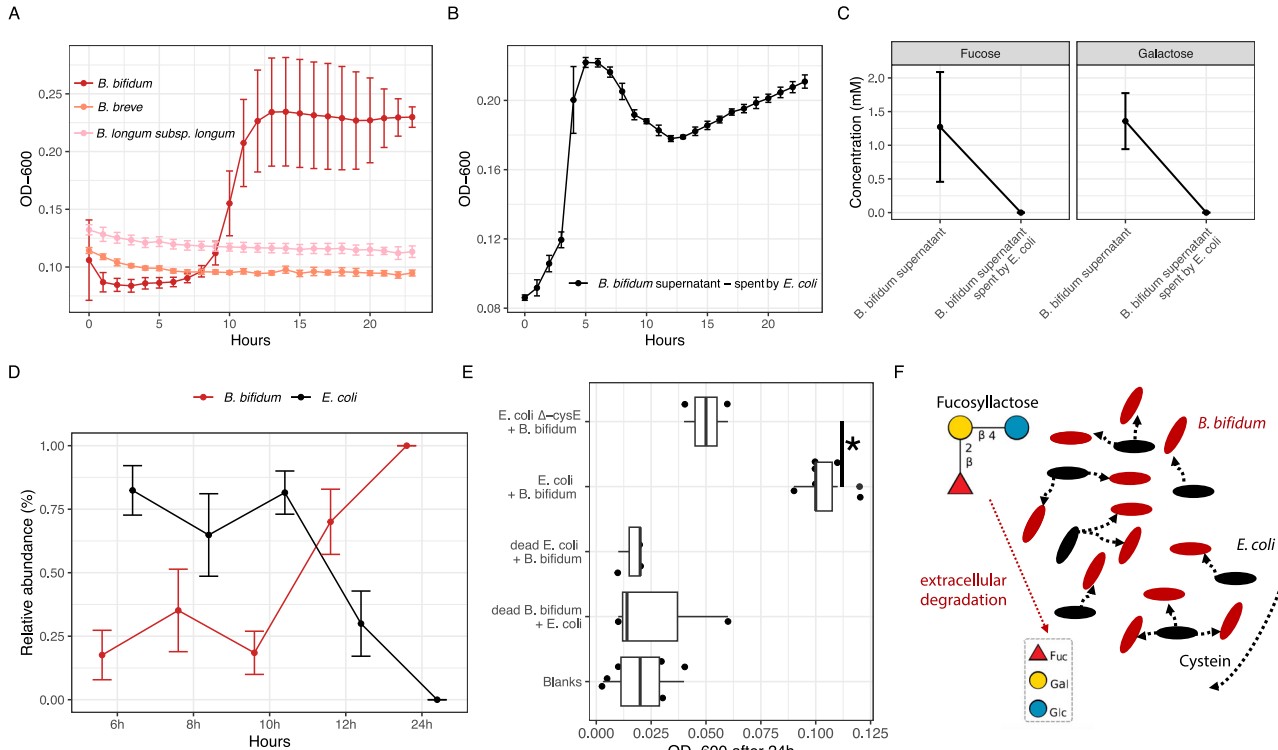

**Fig. 4 | Cooperative degradation of 2′-O-fucosyl-lactose between *E. coli* and *B. bifidum*.** **A** Growth of *B. bifidum* (red), *B. breve* (salmon), *B. longum* subsp. *longum* (pink) in modified M9-medium (m-M9) with 2′-O-fucosyl-lactose (2′FL) as the only source of carbon, measured by optical density (OD-600). Data represent the mean ± s.e.m. of 3 incubation vials per species. **B** Growth of *E. coli* in supernatant *of B. bifidum* previously grown in m-M9 with 2′FL as the only carbon-source. Data represent the mean ± s.e.m. of 6 incubation vials. **C** Concentration of fucose and galactose in m-M9 + 2′FL *B. bifidum* supernatants spent by *E. coli* at pH 6,3. Data represent the mean ± s.e.m. of 10 incubation vials. **D** Relative abundances of *B. bifidum* and *E. coli* in co-culture in m-M9 + 2′FL without any additions of cysteine or other amino acids. Relative abundances were estimated by fluorescent gram-stain and manual counting, averaging 9 fields of view per timepoint. Data represent the

mean ± s.e.m. of 3 co-cultures. **E** Growth of *B. bifidum* and *E. coli* co-cultures with respective controls, including a cysteine mutant (*E. coli* Δ-cysE) which is incapable of cysteine production. Growth was assessed by optical density measurements (OD-600) after 24 h. Blanks included sterile m-M9 + 2′FL medium without cysteine or other amino acids. Boxplots represent the mean ± s.e.m. of 3 to 6 biological replicates. Boxplots show the median (center line), interquartile range (box; 25th–75th percentile), and whiskers extending to the most extreme values within 1.5×IQR. Points represent individual measurements. Differences in OD-600 were assessed using two-sided pairwise Welch's *t*-tests with FDR correction for multiple comparisons (*p*.adj = 0.01). **F** Visual conclusion of the article. Asterisks represent adjusted *p*-values: *$p < 0.05$. Dots in line-plots show group median, error bars show standard deviation. Box plots show group median and interquartile range.

*B. bifidum* cultures, observing growth only when the pH was adjusted to its initial value of 6.3 (Fig. 4B). Repeating the experiments with higher 2′FL concentrations (0.5% w/v) confirmed that *E. coli* could utilize degradation products released by *B. bifidum* (Fig. S5E). Using thin-layer chromatography (TLC) and high-performance anion exchange chromatography (HPAEC) we determined that *B. bifidum* efficiently degraded 2′FL, but left galactose and fucose unconsumed, which were subsequently utilized by *E. coli* K12 (Figs. 4C and S6A–D).

We next hypothesized that under nutrient-limiting conditions, prototrophic *E. coli* K12 could provide essential amino acids to an auxotrophic partner, thus facilitating coexistence. To test this, we co-cultured *E. coli* K12 and our cysteine auxotroph *B. bifidum* isolate in m-M9 + 0.5% 2′FL(v/v) without added cysteine and casamino acids. After 24 hours, we observed turbidity in this co-culture, suggesting growth (Fig. S5F). Fluorescent Gram-staining confirmed co-existence of both species throughout the experiment, until *B. bifidum* became dominant after 12 h (Fig. 4D, and Supplementary Data 5). Furthermore, no significant growth was observed when one partner was ethanol-killed before inoculation, confirming the necessity of active metabolic interaction. Additionally, we repeated the co-culture using an *E. coli* K12 mutant (Δ-cysE) lacking cysteine biosynthesis, observing a significant impairment of growth (*t*.test, *p*.adj = 0.01; Fig. 4E). We therefore conclude that prototrophic *E. coli* K12 can facilitate the growth and metabolic activity of auxotrophic *B. bifidum* under

nutrient-limiting conditions by alleviating its cysteine dependency. In turn, *B. bifidum* enables the degradation of 2′FL, providing *E. coli* with essential carbon sources, demonstrating a reciprocal cross-feeding interaction.

## Discussion

*Bifidobacterium* and *E. coli* are both core members of neonatal gut microbiota[18,49,50]. Here, we examined their co-existence in exclusively breastfed, term-born infants during the initial year post-delivery using strain-resolved metagenomics, a novel computational pipeline (MAJIC), and an annotation-agnostic approach to infer shared ecological constraints during early-life microbiome assembly. We found that *E. coli* frequently occurs alongside one or more highly-abundant *Bifidobacterium* species (*B. longum* subsp. *longum*, *B. breve*, or *B. bifidum*) as long as breastmilk constitutes the primary source of nutrition (Figs. 1 and S1). Because this co-occurrence is not unique to extracellular HMO-degraders, we argue that residual lactose is likely the primary determinant of *E. coli* persistence. However, both genera share ecological constraints related to carbohydrate acquisition in the infant gut, evidenced by their non-random enrichment of f$^{f}_{CAZy}$ (Fig. 3A). Lastly, we demonstrate in vitro that *E. coli* supplies cysteine to auxotrophic *B. bifidum*, enabling joint degradation of 2′FL (Fig. 4F), suggesting that HMO-derived cross-feeding may represent a secondary source of simple sugars for *E. coli*.

Traditionally, interactions between *Bifidobacterium* and *E. coli* in the healthy infant gut had been interpreted as antagonistic, in part due to acidification during HMO fermentation[51,52]. The metabolic bifid-shunt yields acetate and lactate, which can lower environmental pH and are thought to restrict *E. coli* growth[53]. Also, in our batch-culture experiments, *E. coli* was unable to grow in *B. bifidum* spent medium (pH = 4.3) without prior pH adjustment. However, fecal pH in breastfed infants does not fall below 4.3 and has remained above 5.5 for the past decades[54], well above the threshold for lethal intracellular accumulation of undissociated organic acids in *E. coli*. In vivo, luminal pH is further buffered by host bicarbonate secretion[55] and microbial cross-feeding of fermentation end-products[56]. Furthermore, *E. coli* possesses multiple acid tolerance mechanisms, including proton pumps, amino acid decarboxylation systems, and the production of alkaline meta-bolites, which enable survival and colonization under moderately acidic conditions[57]. Here, we highlight that *E. coli* is present in most breastfed infants within our cohort and co-occurs consistently with *Bifidobacterium*, which argues against strict exclusion by acidification in term-born infants and instead points to stable ecological conditions that support *E. coli* at low relative abundance. This raises the question of which resources sustain *E. coli* populations in the healthy infant gut.

Freter's nutrient niche hypothesis offers a plausible explanation: to co-colonize the infant gut, each species must use at least one lim-iting nutrient better than all other species[58,59]. Breastmilk introduces two sources of carbohydrates to colonizing microbiota, HMOs and lactose. *E. coli* is particularly competitive for the acquisition of lactose due to its high-affinity transport system and rapid growth kinetics[27–29]. Since most lactose is absorbed by the host in the small intestine, only limited amounts reach the distal gut[30], likely having a key role in constraining *E. coli* to low relative abundances. In contrast, *Bifido-bacterium* are specialized in degradation of HMOs, which are not absorbed by the host and therefore constitute a large and exclusive carbohydrate pool supporting their high relative abundance[60,61]. Moreover, the extensive structural diversity of HMOs[62] enables niche partitioning among *Bifidobacterium* species, facilitating their stable co-occurrence. Formula-fed infants lack HMOs, diminishing the selective advantage of HMO specialists and favoring metabolism of simple sugars, consistent with higher *E. coli* abundances in these infants[63]. However, our experimental data show that cross-feeding of HMO-degradation products can sustain *E. coli* growth in vitro, suggesting that cross-feeding could act as an additional secondary mechanism influencing *E. coli* abundances in breastfed infants, given the presence of extracellular HMO-degraders such as *B. bifidum*.

On the one hand, HMO cross-feeding interactions may increase substrate use efficiency[64], resulting in enhanced SCFA production[65], a hallmark of a healthy infant gut ecosystem. On the other hand, cross-feeding interactions involve extracellular release of public goods that enlarge niche space but become susceptible to exploitation by pathogens[8,66,67]. These opposing effects raise a critical question: what is the role of cross-feeding for the stability of microbial communities during early-life assembly? Ecological theory predicts that obligate syntrophic interactions are destabilizing, since the mutual depen-dence between cooperating species leads to disruptive positive feed-back loops whereby coupled species either bloom or collapse together, resulting in highly dynamic and unstable microbial communities[67]. This means that, although cooperation can increase the overall metabolic efficiency of microbial communities, it comes at the cost of ecosystem stability. Generally, there are several mechan-isms by which cooperation can be maintained in the gut microbiota in order to optimize these trade-offs. Competitive interactions among other microbial species can dampen 'boom-and-bust' dynamics by limiting resource availability to cooperating partners[68]. Immunological constraints regulate microbial composition[69]. Spatial heterogeneity of the intestinal ecosystem can lead to the separation of interacting species[70,71]. Finally, functional redundancy replaces single strong interactions with multiple weaker ones, while simultaneously buffering communities against loss of metabolic functions[67].

Our in vitro observation of cross-feeding between *E. coli* and *B. bifidum* is an extreme case of obligate mutual dependence, imposed by low-nutrient conditions lacking both easily accessible lactose and cysteine. Conceptually, this interaction is consistent with a snowdrift game, where the success of *E. coli* is maximized under conditions of high enzymatic costs and low capture efficiency by *B. bifidum*, whereas increased nutrient availability would alleviate dependence and shift the mutualistic interaction to either commensalistic behavior or exclusion of one partner[72]. How does this theory relate to early-life microbiota? In the healthy breastfed infant gut, the coupling between *E. coli* and *B. bifidum* is attenuated by multiple stabilizing forces, including host- and breastmilk-mediated immunological constraints, spatial separation of taxa along the developing mucosal landscape, functional redundancy among both genera, and competition arising from residual lactose, a byproduct of incomplete host digestion that may promote stability by increasing competitive overlap among early colonizers. However, in diseased or premature infants, the nutrient and redox landscape is altered[73], lactase function and transit are perturbed[74], and cysteine availability can be limited[75], potentially shifting interactions towards stronger metabolic interdependence among gut microbes. Consistent with this framework, extracellularly feeding *B. bifidum* and *E. coli* are often detected in gut microbiotas of extremely premature infants[76], although this pattern likely reflects multiple concurrent drivers.

Lastly, the "restaurant hypothesis" extends Freter's nutrient niche framework by proposing that cross-feeding interactions expand the available niche space: extracellular degraders act as "restaurants" that release metabolic byproducts, creating new nutritional opportunities for species that cannot access the primary substrate[77,78]. As discussed above, such public goods are inherently susceptible to exploitation, but resident commensal *E. coli* may contribute to colonization resis-tance by limiting access to these nutritional niches. This can occur either through metabolically flexible strains that concurrently utilize residual lactose and HMO-derived monosaccharides[79], or through the coexistence of multiple strains with distinct monosaccharide pre-ferences, both of which reduce carbohydrate availability to other potentially pathogenic *Enterobacteriaceae*. However, while shotgun metagenomics sensitively profiles microbial genomes, without tar-geted qPCR we cannot reliably quantify *Enterobacteriaceae* below ~0.1% relative abundance or evaluate the ecological relevance of ultra-low-abundance strains that fall below our detection limit.

In summary, our data indicate that early-life gut microbiotas are structured primarily by competition for host-derived sugars, while cross-feeding of HMO-degradation products may additionally mod-ulate ecosystem stability and accessibility, given the presence of extracellular HMO-degraders. However, it remains unclear whether cross-feeding interactions originate from evolved traits or passive processes, such as cell lysis or overflow metabolism. Future research should prioritize understanding how microbes interact and compete for key nutrients within their metabolic networks. Stable isotope–based tracing of carbon fluxes across taxa offers a powerful approach to uncover these dynamics, helping guide the rational design of probiotics and/or dietary interventions to support child health.

## Methods

### Experimental model, clinical definitions, stool sample collection

Stool samples were collected within the context of the LucKi Gut study, an ongoing longitudinal study[31] among newborns and their families. Written informed consent was obtained from both parents/legal caregivers prior to enrolment in the study. This research con-firmed to the principles of the Helsinki Declaration. Ethical approval was obtained by the Medical Ethical Committee of Maastricht Uni-versity Medical Center (study number: METC-15-4-237). Pregnant

women residing in the South Limburg region of the Netherlands were recruited through obstetrics and gynecology clinics, lactation information sessions, and advertisements at pregnancy yoga classes, baby clothing stores, and on social media. Infants born prematurely (gestational age ≥32 weeks) were excluded. One maternal sample and 9 infant fecal samples were collected throughout the first 14 months of life for the overarching LucKi study. Participants received fecal sampling starter kits consisting of stool collection tubes (Sarstedt, REF 80.623.022), cold transport containers (Sarstedt, REF 95.1123), safety bags, gloves, faeces collection devices (Fe-Col, REF FC2010), questionnaires, instructions and consent forms. The samples were collected at home and immediately stored at −20 °C in their home freezers. Samples were thereafter transported to the family's well-baby clinic using frozen transport container to preserve the cold chain. From there, samples were transported to the laboratory, where frozen fecal matter was aliquoted and stored at −80 °C until further analyzes.

At each fecal sampling time-point, parents also completed a questionnaire gathering information on the infant's lifestyle, health, development, medication use, and feeding practices, as well as maternal health (during pregnancy), diet, and medication use. Samples collected at 2 weeks (maternal sample), and 2, 6, and 11 months (infant samples) post partum were selected for the purpose of the present study. All selected infants were healthy and born after 38 weeks of gestation and were exclusively breastfed initially.

### Isolation of DNA, metagenomic sequencing, and read-based processing

DNA was extracted using the Power Soil Pro Kit (Qiagen) following the manufacturer's protocol for stool samples, with the inclusion of one negative control per extraction batch, consisting of nuclease-free water to allow for assessment of contamination. DNA was eluted in 40 µl nuclease free water and stored at −20 °C until further analysis. DNA was quantified using Qubit 2.0 dsDNA BR assay kit (ThermoFisher Scientific, Q32850). Library preparation was done following the manufacturer's instructions using the DNA flex kit with 8-mer UDI. Shotgun metagenome sequencing (short-read) on DNA samples was performed at Azenta Life Sciences on Illumina NovaSeq platform at paired-end reads 2× 150 bp. Short-read metagenomic data were processed using HUMAnN 3[80] with default setting. Briefly, HUMAnN 3 first estimates community composition with MetaphlAn 4[81], second it maps reads to a community pangenome with bowtie2[82], and third it aligns unmapped reads to a protein database using DIAMOND[83].

### De novo assembly and processing of metagenome-assembled genomes (MAGs)

Raw sequence reads were quality filtered (-q 20) with fastp v.0.20.0[84] before de novo genome assembly using SPAdes v.3.14.1[85] at default parameters. CheckM v.1.1.3[86] was used for the assessment of metagenome assembled genomes (MAGs). Low-quality genomes (contamination > 20% and completion < 70%) excluded from further analysis. These were taxonomically classified using GTDBtk (v. 2.1.0)[87]. The MAG dataset was dereplicated with standard settings via dRep[88], resulting in a final set of 458 unique quality MAGs (Supplementary Data 3). Dereplicated MAGs were furthermore concatenated into a single fasta file, and bowtie2[82] was used to create a mapping index from it, as well as to calculate the percentage of reads per sample that were mapping to our set of dereplicated MAGs. A "scaffold-to-bin" was created using "parse_stb.py" from dRep[88], and Prodigal[89] was used to profile all genes from the concatenated MAG file, thereby creating a "gene" file. Combining these files, "inStrain profile" was used to screen within-sample nucleotide diversity (π) and gene-nucleotide diversity (gene-π), and "inStrain compare" was used to profile across-sample population average nucleotide identity (popANI)[41]. The 'gene' file was furthermore used for annotation of all genes using EggNOG mapper v. 6.0[42].

### Detection of glycoside-hydrolase-linked genes

We expanded a previous analysis that detected genes that coevolve with chitinases[44] by analyzing 21,559 high-quality species-representative genomes from GTDB r214 (>99% completeness, <1% contamination). Genes were called with Pyrodigal[90], and full-length proteins were clustered at 70% similarity using diamond deepclust followed by diamond recluster (both with --approx-id 70)[83]. Partial reading frames were then added to clusters using diamond blastp (--query-cover 100 --id 80 -b 20 -c 1 -k 10). Glycosyl hydrolases were called with dbcan4[91]. All genes were annotated using eggnog-mapper v5[92]. To look for genes that co-evolve with GH2, we only considered protein clusters that appear in >9 genomes, which reduces computational load and false positive rates due to random fluctuations[44]. Ancestral states were reconstructed using maximum parsimony, as implemented in the mpr function of the PhyloTools Julia package[93]. We detected co-evolution as correlated gain-loss events using canonical correlation analysis (CCA). We used CCA because there are many GH2 protein clusters, and CCA uncovers how the optimal linear combination of their gain/loss vectors correlates with a target vector. These correlations were compared to null distributions generated from randomized vectors, where events from different genes were randomized, therefore controlling for the mean tendency of nodes to display gain/loss events[44]. To analyze MAGs that were assembled de novo from our own data, we first assign MAG proteins to the protein homology clusters defined above using diamond blastp (--query-cover 80 --subject-cover 80 -k 3 -c 1). Each MAG protein was assigned to the best-matching protein cluster based on e-value. The number predicted number of GH2 enzymes in each MAG was calculated using the random forest regression model from above, and compared to the observed number of GH2 proteins encoded by the MAG. To identify the fraction of GH2 co-evolved genes that themselves are carbohydrate-active enzymes, we compared annotated coevolving genes against the PFAM database using PFAM.db v3.8.2[94].

### Growth assays with *Bifidobacterium* and *E. coli*

*B. bifidum*, *B. longum* subsp. *longum*, and *B. breve* were isolated from infant stool samples by spreading 10⁻⁷ dilutions of fecal slurries onto MRS agar with cysteine (0.1% w/v; Sigma-Aldrich) and mupirocin (50 mg/l; Sigma-Aldrich) (m-MRS). Plates were incubated at 37 °C under anaerobic conditions for 48 h. Per plate, several colonies were picked at random and grown in liquid m-MRS, repeating the 48 h-long incubation. From incubations with visible growth, DNA was extracted using the Power Soil Pro Kit (Qiagen) following the manufacturer's protocol and finally eluted in 40 µl nuclease-free water. DNA isolated from pure bacterial cultures was subjected to whole-genome sequencing via Illumina NextSeq 2000. Obtained sequencing reads were pre-processed with fastp v.0.23.2[84]. Unicycler v.0.4.9[95] with the "–mode conservative" option was used to produce assemblies, after which contigs below 1000 bp were filtered out. Genome completeness and contamination were estimated using CheckM v.1.2.0[86], and we retained sequences with completeness >99% and contamination < 1%. GTDB-Tk v.2.1.0[96] was used to classify assemblies to the strain level. Thereby, we identified several *B. bifidum*, *B. longum* subsp. *longum*, and *B. breve*, and selected one representative respectively for our growth assays.

*Bifidobacterium* isolates were pre-grown in MRS-medium + cysteine (0.1% w/v; Sigma-Aldrich) for 48 h. Thereof, 2% (v/v) was transferred to modified M9 medium[47] with 0.1% or 0.5% 2'-O-fucosyl-lactose (2'FL; Glycome, IRE), following a 24 h incubation at 37 °C under anaerobic conditions (Coy Laboratory Products, USA) to probe their capabilities of metabolizing 2'FL. Modified M9 had the following additions: biotin (10 mg/l), para-aminobenzoic acid (10 mg/l), thiamine (400 mg/l), nicotinic acid (400 mg/l), pyridoxine (400 mg/l), pantothenate calcium (200 mg/l), riboflavin (200 mg/l), and a diluted casamino-acid extract (0.01% w/v; Sigma-Aldrich). pH was adjusted to

6.3, and media were filter-sterilized (Sartorius 0.22 μm filter; Sigma-Aldrich) before inoculation. *E. coli* K12 was pre-grown in brain-heart infusion (BHI; Sigma Aldrich) for 24 h before being inoculated (1% v/v) in m-M9 or m-M9 spent by *B. bifidum*. To generate supernatants of spent media, cultivates were centrifuged at 13,000 rpm for 10 min, supernatants were collected, pH was adjusted to 6.3, and lastly filter-sterilized (Sartorius 0.22 μm filter; Sigma-Aldrich). Growth of all cultures was monitored under anaerobic conditions in a plate-reader (Thermo Multiskan, Thermo Fisher, USA). For end-point assessments, optical density (OD-600) of cultures was measured by spectro-photometry (Camspec, Spectronic, UK). *E. coli* BW25113 Δ-cysE was obtained from the Keio collection[97]. For co-culture experiments, *B. bifidum* and *E. coli* K12 were pre-grown in m-MRS or BHI for 48 or 24 h under anaerobic conditions at 37 °C, respectively. Cell density of inoculates was assessed by Luna FX7 automated cell counter (Thermo Fisher USA) to ensure close to 1:1 ratios of all inoculates. *E. coli* Δ-cysE was pre-grown in BHI + cysteine. Relative abundances of *B. bifidum* and *E. coli* K12 in co-culture were assessed by fluorescent Gram stain (LIVE BacLight, Thermo Fisher, USA) and manual counting. Nine representative pictures were taken per timepoint (Zeiss, GER) to estimate the average relative abundance of either microbe in co-culture (Supplementary File 1).

### Quantification of 2′-O-fucosyl-lactose and its monomers

For TLC, Culture supernatant samples (3 μL) were spotted onto silica plates (Sigma Z740230) and resolved in running buffer containing 2:1:1 butanol, acetic acid and dH$_2$O, respectively. TLCs were subsequently stained using diphenylamine–aniline–phosphoric acid stain[98] and developed by heating.

To analyze and quantify monosaccharide release in culture supernatants, we used high-performance anion exchange chromatography with pulsed amperometric detection (HPAEC-PAD). Sugars were separated using a CarboPac PA-1 anion exchange column with a PA-1 guard using a Dionex ICS-6000 (Thermo Fisher) and detected using PAD. Flow was 0.75 mL min$^{-1}$ and elution conditions were 0–25 min 5 mM NaOH and then 25–40 min 5–100 mM NaOH. Software used was the Chromeleon Chromatography Data System. Mono-saccharide/disaccharide standards were included at the following concentrations: Maltose = 0.1 mg/ml, Glucose = 0.1 mM, Fucose = 0.1, 0.075, 0.05, 0.025, and 0.01 mM, N-acetylglucosamine = 0.2 mM, Galactose = 0.2 mM, N-acetylgalactosamin = 0.1 mM, and Lactose 0.2 mM. All data were obtained by diluting supernatant samples 1/10 prior to injection.

### Statistics, reproducibility, and data visualization

Statistical analysis (Student's *T*-test, ANOVA, repeated measures ANOVA, Wilcoxon Test, Chi-square test) was performed in "R version 4.0" and the R package "rstatix version 0.7.0"[99]. All *p*-values were adjusted using Bonferroni's method. Data was visualized via "R version 4.0" and R package "ggplot2 version 3.3.340"[100]. "Ampvis2"[101] and "phyloseq"[102] were used for handling of metagenomic counts, meta-data, and taxonomy files, including species filtering, analysis of alpha and beta diversity, as well as PCA. "co_occurrence()" in phylosmith[103] was used for co-occurrence analysis of microbial species. For randomization of matrices, we used the "randomizeMatrix()" function in "Picante"[104] with "null.model = "frequency" over 10,000 iterations. Kolmogorow–Smirnov-Test – "ks.test()" in base R – was used to test for whether differences in randomized and observed matrices were statistically different. "c.score()" function in "bipartite"[105] was used for calculation of checkerboard indices. For MAJIC, randomized and observed matrices were transformed into presence/absence matrices using "phyloseq_standardize_otu_abundance()" with "method = pa" in "metagMisc"[106], and average Jaccard dissimilarities as well as cross-sample Shannon diversity (CSI) index per species were calculated using the functions "vegdist()" and "diversity()" in "vegan"[107],

respectively. MAJIC was coded "R version 4.0", looping through focal species $\sigma_i$ to obtain $\sigma_i^+$ ($\mu_i^+$), and in $\sigma_i^-$ ($\mu_i^-$) - as described in Fig. S2A.

## Data availability

Source data are provided with this paper. Further information for resources and reagents should be directed to and will be fulfilled by lead contacts David Seki (david.seki@univie.ac.at) or Lindsay J. Hall (l.hall.3@bham.ac.uk). Infant fecal sample metagenome sequencing raw reads are publicly available in the NCBI Sequence Read Archive (SRA) under accession no. PRJNA1230889. All 458 MAGs recovered from gut metagenome samples are available via Zenodo[108] (https://doi.org/10.5281/zenodo.18848062), and all high-quality MAGs with completeness > 95% (n = 279) are also available via PRJNA1230889. Source data are provided with this paper.

## Code availability

The code for analysis of the data[109] can be accessed via https://github.com/sekid666/MAJIC and https://doi.org/10.5281/zenodo.18833415.

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

## Acknowledgements

We thank all families for their participation during this study. We would like to thank the QIB sequencing team for technical assistance with shotgun metagenomics library preparation. This project was funded by the Wellcome Trust Investigator Award no. 220876/Z/20/Z to L.J.H., and a Biotechnology and Biological Sciences Research Council (BBSRC) Institute Strategic Programme, Gut Microbes and Health BB/R012490/1, and its constituent projects BBS/E/F/000PR10353 and BBS/E/F/000PR10356, and by the BBSRC Institute Strategic Programme Food Microbiome and Health BB/X011054/1 and its constituent project BBS/E/F/000PR13631 to L.J.H. D.S. and S.P. acknowledges funding from the Austrian Science Fund (FWF; doi.org/10.55776/COE7).

## Author contributions

Conceptualization, D.S. and L.J.H; methodology, D.S., M.K., R.K., S.P., L.I.C., C.R.B., A.AG.; software, D.S., S.P; investigation, D.S.; writing – original draft, D.S. and L.J.H.; writing – review & editing, D.S., L.J.H., S.P.; resources, L.J.H., L.I.C., N.v.B., J.P., P.T.C., M.M.; and funding acquisition, L.J.H.

## Competing interests

The authors declare no competing interests.
