## [Transparent Peer Review file · Nature Communications]

Human milk oligosaccharide mediates mutualism between *Escherichia coli* and *Bifidobacterium bifidum*

Corresponding Author: Professor Lindsay Hall

Version 0:

Reviewer comments:

Reviewer #1

(Remarks to the Author)

Summary

Seki et al., analyse the gut microbiome of infants in a Dutch longitudinal cohort (2m, 6m and 11m samples) as well as maternal gut microbiomes. The authors identify co-occurrence between *E. coli* and dominant *Bifidobacterium* species in infant samples from 2m and 6m (during breastfeeding, when the main carbohydrate source for gut microbes is HMOs). Subsequently, they show, in vitro that *Bifidobacterium bifidum* liberates Fucose and Galactose from 2-FL that is then cross-fed to *E. coli*, while this species provides cysteine to *B. bifidum* (a cysteine auxotroph), which may explain the observed co-occurrence of the two species in the infant cohort. There are, however, some major issues with the present study. While the in vitro studies elegantly show mutualism between *E. coli* and *B. bifidum*, the relevance of this to explain observations in the infant gut ecosystem is in my opinion highly doubtful.

Major concerns

- 1) The authors claim that *E. coli* presence in the infant gut depends on having a major dominating HMO-utilizing species (in most cases a *Bifidobacterium* species). However, *E. coli* is very prevalent in the cohort (76-96%) and the authors use a cutoff for presence/absence of 0.1% relative abundance. It is not unlikely that *E. coli* is in fact present in all samples but either below the selected cutoff or just right below the limit of detection of sequencing. Without more sensitive methods to determine presence/absence (e.g. qPCR using *E. coli* specific primers), I am not confident that the author's conclusion holds true. Furthermore, the authors group all "Bifidobacterium dominant" samples (Fig. 1E), without stratification into specific *Bifidobacterium* species, yet they only focus on *B. bifidum* for the later in vitro experiments (see point below).
- 2) Based on the human gut microbiome data, the authors suggest that HMO-utilizing *Bifidobacterium* species cross-feed HMO-degradants to *E. coli*. Yet, this is very unlikely to be true for other *Bifidobacterium* species than *B. bifidum*, since *B. longum* and *B. breve* (in contrast to *B. bifidum*) strains mostly employ intracellular HMO-degradation strategies (PMID: 31888048) and therefore will not liberate and cross-feed HMO sugars to *E. coli*. This is a problem and creates a disconnection between the human data and the in vitro data.
- 3) The authors claim (l. 37-39) that *E. coli* relies on primary HMO degraders to persist in the gut, yet they do not include any formula-fed infants as controls AND previous studies show that *E. coli* prevalence and abundance is in fact even higher in formula fed infants (PMID: 15668012).

Minor comments

L. 124-133. I don't understand this way of using Shannon index. Shannon index tells you something about species diversity in a sample – it doesn't make sense to do for individual species? Are you looking at Shannon index of all species in samples where the indicated species are present?

L. 134-140. How can you rule out that all samples in fact have *E. coli* present (but just at rel abundance below <0.1%)? Is the statistical analysis based on both 2m and 6m data? What happen if you include only 2m or 6m? What about 11m?

L. 138-143. I'm not sure I agree that 20% rel abundance equals dominance. In my view this would mean >50%. I assume that using this definition of dominance (>20%), implies that no other bacterial species in the sample reached higher rel abundance? Please elaborate.

Fig. 1. Panel B. Chao-1 estimates richness, so I guess it is not "observed diversity"? Did you look at actual Observed species (at a certain seq depth cutoff) and/or Shannon index? What about evenness? Chao estimated richness seems quite

low, especially considering maternal samples with only 20-40 species?

(Remarks on code availability)

Reviewer #2

(Remarks to the Author)

The manuscript by Seki and colleagues elegantly builds upon the mechanistic understanding of the coexistence between Enterobacteriaceae and Bifidobacterium in the breastfed infant gut. The study presents a coherent and logical progression from an extensive set of computational analyses across multiple metagenomes, including the use of newly developed custom tools, to experimental validation of one of the molecular mechanisms underlying the coexistence of *E. coli* and *B. bifidum*. Specifically, the authors demonstrate a cooperative degradation of 2' fucosyllactose, initiated by Bifidobacterium, coupled with the provision of cysteine by *E. coli* to coexisting bifidobacteria.

The computational analyses begin by defining the dominant and coexisting taxa in the infant gut, assessing strain sharing between mothers and infants, and characterizing the prevalence of specific strains over time. Notably, the authors report that *E. coli* consistently coexist with species capable of utilizing human milk oligosaccharides (HMOs), leading to the hypothesis of a cooperative relationship between these taxa. Furthermore, through the integration of existing analytical tools and the development of novel computational methods, the study investigates in silico how pairs of strains influence each others' abundance and ecological dynamics.

Both the methodological framework and the biological insights presented are innovative and significant. The work introduces new computational approaches for microbial ecology, provides fresh evidence supporting distinct mechanisms of microbial transmission and persistence in the infant gut, and identifies putative interacting microbial pairs. Most notably, the authors construct and in vitro validate a compelling hypothesis that *E. coli* and Bifidobacterium bifidum cooperate through resources sharing in the infant gut: *B. bifidum* acts as the primary degrader of HMOs, generating substrates that support *E. coli* growth, while *E. coli* reciprocally provides cysteine to the anaerobic Bifidobacterium.

Other comments:

According to cohort description, last infant sample included in the investigation was collected at 11 months however, demographic characteristics in Table 1 compile characteristics at 14 months, but not at 11. Please revise if there is any mistake.

The interpretation of loose and restricted genes according to their gene-pi values is not very clear, do the authors hypothesized a distinction in this parameter among genes required for host adaptation as opposed to the ones "dispensable" genes for this adaptation? There seems to be a wide variety of function categories equally distributed among both categories so the interpretation of the meaning of this parameter is not clear.

In relation to the growth experiments, the growth of *B. bifidum* is very limited according to the growth curves provided in Fig S5, that also present important standard deviations. Do the authors have data on cfu counts or other parameters to demonstrate the slight increase in OD is actually due to effective growth and not to media components sedimentation?

In relation to information provided in line 355, I think it is well established that bif requires cysteine supplementation for proper growth in culture, even in richer media

Lines 372-374 and supplementary fig 5F: I am not sure that this image is actually showing robust coculture growth, as stated, from the image it is impossible to know if both strains are actually growing or if the turbidity is just due to the growth of one of them.

Lines 468: were excluded infants born at gestational age ≥ 32 weeks, or those ≤ 32 weeks? Revise.

(Remarks on code availability)

Version 1:

Reviewer comments:

Reviewer #1

(Remarks to the Author)

See attached pdf.

(Remarks on code availability)

Reviewer #1 (Remarks to the Author):

Summary

Seki et al., analyse the gut microbiome of infants in a Dutch longitudinal cohort (2m, 6m and 11m samples) as well as maternal gut microbiomes. The authors identify co-occurrence between *E. coli* and dominant Bifidobacterium species in infant samples from 2m and 6m (during breastfeeding, when the main carbohydrate source for gut microbes is HMOs). Subsequently, they show, in vitro that Bifidobacterium bifidum liberates Fucose and Galactose from 2-FL that is then cross-fed to *E. coli*, while this species provides cysteine to *B. bifidum* (a cysteine auxotroph), which may explain the observed co-occurrence of the two species in the infant cohort. There are, however, some major issues with the present study. While the in vitro studies elegantly show mutualism between *E. coli* and *B. bifidum*, the relevance of this to explain observations in the infant gut ecosystem is in my opinion highly doubtful.

Major concerns

1) The authors claim that *E. coli* presence in the infant gut depends on having a major dominating HMO-utilizing species (in most cases a Bifidobacterium species). However, *E. coli* is very prevalent in the cohort (76-96%) and the authors use a cutoff for presence/absence of 0.1% relative abundance. It is not unlikely that *E. coli* is in fact present in all samples but either below the selected cutoff or just right below the limit of detection of sequencing. Without more sensitive methods to determine presence/absence (e.g. qPCR using *E. coli* specific primers), I am not confident that the author's conclusion holds true.

We thank the reviewer for this thoughtful comment. While it is true that *E. coli* might be present in samples below our detection threshold, we consider that presence at such low abundances (<0.1%) would ecologically be negligible. Cross-feeding interactions require sufficient biomass to generate metabolite concentrations that would sustain growth of both cross-feeding partners. Thus, populations below this level are unlikely to participate meaningfully in community dynamics. We have clarified this in the revised manuscript.

We appreciate the reviewer's concern, and we conducted additional analyses in the revised manuscript in response, aiming to underscore the validity of our approach [revised manuscript lines 220-228 and revised Figure S3]. Unfortunately, we no longer have access to the original samples for qPCR quantification (as remaining samples were destroyed due to ethical/MTA requirements), and the limited archived DNA we hold, stored at -20 °C, is unlikely to yield reliable results. However, we emphasize that shotgun metagenomics provides a highly sensitive and quantitative approach for detecting bacterial genomes. To assess whether sequencing depth was sufficient for confident detection, we performed a rarefaction analysis in the revised manuscript using the breadth (fraction of genome covered by reads) and coverage (average read depth) of assembled MAGs. This approach directly evaluates genome detection completeness and avoids biases from marker-gene-based taxonomic profiling (e.g., MetaPhlAn). In this context, coverage serves as a proxy for sampling effort, while breadth serves as a measure of genome detection (<https://doi.org/10.1038/s41587-020->

00797-0). Plotting breadth against coverage allows us to evaluate whether increasing sequencing depth would yield additional genomic information.

Plotting breadth against coverage revealed a knee at ~ 0.34 and a plateau near ~ 0.85 , indicating that additional sequencing would yield minimal new genomic information. This suggests that sequencing depth was sufficient to capture most detectable genomes, including low-abundance taxa with recommended thresholds of breadth > 0.5 and coverage > 5 (as discussed in <https://doi.org/10.1038/s41587-020-00797-0>)

Furthermore, to address the reviewer's concern, we performed a sensitivity analysis to explore the impact of varying thresholds. Using the recommended thresholds (breadth > 0.5 and coverage > 5), our analysis recapitulates the read-based findings, indicating that *E. coli* is highly likely to be detected in infants at both two and six months post-delivery.

To further evaluate the robustness of our criteria, we performed a sensitivity analysis across a range of thresholds (breadth = 0.1–1.0; coverage = 0–20). *E. coli* was consistently detected across samples up to the plateau, particularly at two and six months post-delivery. Detection only declined above ~0.85 breadth, which likely reflects incomplete MAG assembly or uneven coverage rather than true absence. This analysis supports our approach, which balances sensitivity and specificity, capturing ecologically relevant populations while avoiding spurious detections.

Similarly, our 0.1% relative abundance cutoff in the read-based analysis was chosen to exclude ecologically insignificant background signals. Although *E. coli* could theoretically occur below this threshold, such minimal abundances are unlikely to support cross-feeding or other functional interactions. The combination of read-based and genome-resolved evidence, together with the observed coverage-breadth plateau, strongly indicates that sequencing depth was adequate for robust detection of *E. coli* and its ecological associations with HMO-utilizing species.

Finally, while we agree with the reviewer that open questions remain regarding the *in vivo* relevance of the observed mutualism, we believe that additional qPCR screening for very rare *E. coli* would not substantially advance the ecological interpretation. Direct validation of metabolic exchange *in vivo* would instead require stable isotope-labeled HMO experiments to trace carbon flux across taxa. This is technically demanding and highly expensive, but a promising direction for future work. We have clarified these limitations and the rationale for our abundance thresholds in the revised results as

described above, and in our revised discussion (revised manuscript lines: 438-444 and 477-480).

Furthermore, the authors group all “Bifidobacterium dominant” samples (Fig. 1E), without stratification into specific Bifidobacterium species, yet they only focus on *B. bifidum* for the later in vitro experiments (see point below).

2) Based on the human gut microbiome data, the authors suggest that HMO-utilizing Bifidobacterium species cross-feed HMO-degradants to *E. coli*. Yet, this is very unlikely to be true for other Bifidobacterium species than *B. bifidum*, since *B. longum* and *B. breve* (in contrast to *B. bifidum*) strains mostly employ intracellular HMO-degradation strategies (PMID: 31888048) and therefore will not liberate and cross-feed HMO sugars to *E. coli*. This is a problem and creates a disconnection between the human data and the in vitro data.

We thank the reviewer for this useful comment. In the revised manuscript, we clarify the rationale for focusing on *Bifidobacterium bifidum* as described below.

Although *B. bifidum* is the best-characterized extracellular HMO degrader, recent studies indicate that certain *B. breve* and *B. longum* strains can also employ extracellular or semi-extracellular degradation mechanisms (doi: 10.1093/femsre/fuad056 and <https://doi.org/10.1038/srep38560>). Thus, potential cross-feeding with *E. coli* may not be exclusive to *B. bifidum*. We have expanded the discussion in the revised manuscript to reflect this nuance and to highlight that our findings with *B. bifidum* represent one well-resolved example within a broader ecological context.

We focused our *in vitro* validation on *B. bifidum* because it displayed distinct ecological and genomic signatures in our dataset. Specifically, (i) significantly lower gene-level nucleotide diversity compared to *B. breve* and *B. longum* (Fig. 2C), and (ii) GH2-associated genomic traits consistent with extracellular specialization (Fig. 3A). Furthermore, *B. bifidum* was the only species that consistently grew in minimal medium using 2'FL as the sole carbon source. Establishing these conditions was technically challenging but a critical prerequisite for mechanistic testing.

Given these findings, potential cross-feeding interactions between *E. coli* and these bifidobacterial species may not be exclusive to *B. bifidum* and warrant further investigation. While exploring these interactions is beyond the scope of the current study, we acknowledge the importance of this avenue for future research. We have expanded the discussion in the revised manuscript to address this point and hope the reviewer finds our perspective reasonable. Specifically, we highlight now that our data demonstrates one clear example of mutualistic interaction between *B. bifidum* and *E. coli*, while leaving open the possibility that other *Bifidobacterium* species may participate in analogous interactions. (revised manuscript lines: 411-421)

Additionally, in response to the reviewer’s suggestion, we have now included a species-level stratification of *Bifidobacterium* taxa exceeding 20% relative abundance per

sample (revised Supplementary Fig. 1). This analysis reveals that *B. bifidum*, *B. longum* subsp. *longum*, and *B. breve* each contribute to the observed co-occurrence patterns with *E. coli*, albeit to varying degrees.

3) The authors claim (l. 37-39) that *E. coli* relies on primary HMO degraders to persist in the gut, yet they do not include any formula-fed infants as controls AND previous studies show that *E. coli* prevalence and abundance is in fact even higher in formula fed infants (PMID: 15668012).

We thank the reviewer for this comment and for citing the relevant work, which includes one of our co-authors. It is correct that *E. coli* prevalence and abundance can be higher in formula-fed infants. This difference likely arises because formula contains higher levels of free disaccharides, primarily lactose, which *E. coli* can utilize directly. In this context, *E. coli* and bifidobacteria compete for readily available sugars, and primary HMO degradation is not required. In contrast, in breastfed infants, *E. coli* relies on substrates liberated by HMO-degrading bifidobacteria, consistent with our findings. We have clarified this further in our discussion in the revised manuscript. Furthermore, we note that we deliberately focused exclusively on healthy, breastfed infants, as such cohorts are rare and particularly valuable for studying HMO-driven microbial interactions. We have extended the discussion to highlight this point in the revised manuscript lines 447-449.

Minor comments

L. 124-133. I don't understand this way of using Shannon index. Shannon index tells you something about species diversity in a sample – it doesn't make sense to do for

individual species? Are you looking at Shannon index of all species in samples where the indicated species are present?

We thank the reviewer for pointing this out. We agree that the description in the original text was unclear. We did not compute a Shannon index per species within a sample. What we did was to calculate the Shannon diversity of each species' abundance distribution across all samples. This measure reflects how evenly, or consistently a species occurs across individuals in cohort. And we did this to figure out which species are the most consistently occurring. We have revised this in the manuscript accordingly. (revised manuscript lines 120-128)

L. 134-140. How can you rule out that all samples in fact have *E. coli* present (but just at rel abundance below <0.1%)? Is the statistical analysis based on both 2m and 6m data? What happen if you include only 2m or 6m? What about 11m?

We thank the reviewer for pointing this out. Double checking this has improved our manuscript. In fact, we did not separate this analysis by age groups, which was an oversight. We now calculate a Fisher's exact test only using samples from infants B-02, B-06, and B-11 (previously maternal ones were included). Also, we have conducted a sensitivity analysis showing that within any threshold filter between 0-1% relative abundance, this relationship between *E. coli* and bifidobacteria remains significant, therefore stable. (revised manuscript lines 131-136)

L. 138-143. I'm not sure I agree that 20% rel abundance equals dominance. In my view this would mean >50%. I assume that using this definition of dominance (>20%), implies that no other bacterial species in the sample reached higher rel abundance? Please elaborate.

We agree with the reviewer's concern here and will rephrase dominance to 'highly abundant (> 20%)'

Fig. 1. Panel B. Chao-1 estimates richness, so I guess it is not "observed diversity"? Did you look at actual Observed species (at a certain seq depth cutoff) and/or Shannon index? What about evenness? Chao estimated richness seems quite low, especially considering maternal samples with only 20-40 species?

We thank the reviewer for this comment. The y-axis label in the original version of Fig. 1B was incorrect. We have now corrected this panel and replaced it with Shannon diversity, capturing both richness and evenness. Additionally, we have added a rank-abundance curve in the revised manuscript to better visualize the observed diversity in our sample set of infants and mothers. (revised manuscript lines 102-104 and revised Figure S1A)

Reviewer #2 (Remarks to the Author):

The manuscript by Seki and colleagues elegantly builds upon the mechanistic understanding of the coexistence between Enterobacteriaceae and Bifidobacterium in the breastfed infant gut. The study presents a coherent and logical progression- from an extensive set of computational analyses across multiple metagenomes, including the use of newly developed custom tools, to experimental validation of one of the molecular mechanisms underlying the coexistence of E. coli and B. bifidum. Specifically, the authors demonstrate a cooperative degradation of 2´ fucosyllactose, initiated by Bifidobacterium, coupled with the provision of cysteine by E coli to coexisting bifidobacteria.

The computational analyses begin by defining the dominant and coexisting taxa in the infant gut, assessing strain sharing between mothers and infants, and characterizing the prevalence of specific strains over time. Notably, the authors report that E. coli consistently coexist with species capable of utilizing human milk oligosaccharides (HMOs), leading to the hypothesis of a cooperative relationship between these taxa. Furthermore, through the integration of existing analytical tools and the development of novel computational methods, the study investigates in silico how pairs of strains influence each others' abundance and ecological dynamics.

Both the methodological framework and the biological insights presneted are innovative and significant. The work introduces new computational approaches for microbial ecology, provides fresh evidence supporting distinct mechanisms of microbial transmission and persistence in the infant gut, and identifies putative interacting microbial pairs.

Most notably, the authors construct and in vitro validate a compelling hypothesis that E. coli and Bifidobacterium bifidum cooperate through resources sharing in the infant gut: B. bifidum acts as the primary degrader of HMOs, generating substrates that support E. coli growth, while E. coli reciprocally provides cysteine to the anaerobic Bifidobacterium.

We thank the reviewer for their positive and constructive assessment of our study, and we are grateful for the encouraging remarks.

Other comments:

According to cohort description, last infant sample included in the investigation was collected at 11 months however, demographic characteristics in Table 1 compile characteristics at 14 months, but not at 11. Please revise if there is any mistake.

In the entire Lucki cohort, we collected information and samples at 1-2 weeks, 4 weeks, 8 weeks, 4, 5, 6, 9, 11 and 14 months. At 6 months and 14 months we had more extensive questionnaires with information on for example day care attendance. For this study, we only focused on samples from healthy infants, born after 38 weeks of gestation, that were also exclusively breastfed. So, the last stool samples used for this

study were those collected at 11 months, but we did have additional follow-up information. We have clarified in more detail in the revised manuscript. (revised manuscript lines 488-490 and 500-503)

The interpretation of loose and restricted genes according to their gene-pi values is not very clear, do the authors hypothesized a distinction in this parameter among genes required for host adaptation as opposed to the ones "dispensable" genes for this adaptation? There seems to be a wide variety of function categories equally distributed among both categories so the interpretation of the meaning of this parameter is not clear.

We appreciate this comment - it's insightful. At present, we don't have detailed knowledge about the distribution of functional groups among 'loose' and 'restricted' genes. After re-visiting this point, we also realized that we don't have a specific hypothesis regarding which genes are essential for host adaptation and which might be dispensable. In the revised results (and corresponding methods section), we now explicitly state that these categories are operational definitions intended to identify genes with distinct evolutionary dynamics. (revised manuscript lines: 276-283)

In relation to the growth experiments, the growth of *B. bifidum* is very limited according to the growth curves provided in Fig S5, that also present important standard deviations. Do the authors have data on cfu counts or other parameters to demonstrate the slight increase in OD is actually due to effective growth and not to media components sedimentation?

For *B. bifidum* in particular, this could be due to clumping / floc formation in liquid culture. To address the reviewer's concern and to confirm that the observed OD increase corresponds to actual cell growth rather than sedimentation of media components, we repeated the incubation of *B. bifidum* in minimal medium with 2'FL as the sole carbon source. Samples were taken after 3 and 24 hours (in three biological replicates), stained with SYBR-Green, and analyzed by flow cytometry to quantify cell numbers. Below, we show cytometry images of representative replicates, and our data confirmed an increase in cell counts over time, supporting that the OD changes reflect genuine bacterial growth.

B. bifidum 3 hours in m-M9 + 2'FL (0.5%)	B. bifidum 24 hours in m-M9 + 2'FL (0.5%)
Replicate 1: 942,98 cells/ μ l	Replicate 1: 10421,55 cells/ μ l
Replicate 2: 907,74 cells/ μ l	Replicate 2: 8855,57 cells/ μ l
Replicate 3: 928,41 cells/ μ l	Replicate 3: 9479,06 cells/ μ l

In relation to information provided in line 355, I think it is well established that bif requires cysteine supplementation for proper growth in culture, even in richer media

We have highlighted in the revised manuscript that *B. bifidum*'s cysteine dependency is a well-established fact. (revised manuscript lines 357-359)

Lines 372-374 and supplementary fig 5F: I am not sure that this image is actually showing robust coculture growth, as stated, from the image it is impossible to know if both strains are actually growing or if the turbidity is just due to the growth of one of them.

We appreciate this observation. Supplementary Fig. 5F was our initial observation of turbidity in a medium without cysteine or other added amino acids, after being inoculated with *E. coli* and *B. bifidum*. In the revised manuscript, we now state that we observed turbidity. (revised manuscript lines 376-377)

Lines 468: were excluded infants born at gestational age ≥ 32 weeks, or those ≤ 32 weeks? Revise.

This was a description of the whole Lucki cohort. We have now deleted this sentence in the revised manuscript. It is not relevant for the present study, since we have a subgroup of the Lucki cohort (infants selected >38 weeks gestation). This has been clarified in the revised manuscript. (revised manuscript lines 502-503)

Major concerns

1) The authors claim that *E. coli* presence in the infant gut depends on having a major dominating HMO-utilizing species (in most cases a Bifidobacterium species). However, *E. coli* is very prevalent in the cohort (76-96%) and the authors use a cutoff for presence/absence of 0.1% relative abundance. It is not unlikely that *E. coli* is in fact present in all samples but either below the selected cutoff or just right below the limit of detection of sequencing. Without more sensitive methods to determine presence/absence (e.g. qPCR using *E. coli* specific primers), I am not confident that the author's conclusion holds true.

We thank the reviewer for this thoughtful comment. While it is true that *E. coli* might be present in samples below our detection threshold, we consider that presence at such low abundances (<0.1%) would ecologically be negligible. Cross-feeding interactions require sufficient biomass to generate metabolite concentrations that would sustain growth of both cross-feeding partners. Thus, populations below this level are unlikely to participate meaningfully in community dynamics. We have clarified this in the revised manuscript.

We appreciate the reviewer's concern, and we conducted additional analyses in the revised manuscript in response, aiming to underscore the validity of our approach [revised manuscript lines 220-228 and revised Figure S3]. Unfortunately, we no longer have access to the original samples for qPCR quantification (as remaining samples were destroyed due to ethical/MTA requirements), and the limited archived DNA we hold, stored at -20°C , is unlikely to yield reliable results. However, we emphasize that shotgun metagenomics provides a highly sensitive and quantitative approach for detecting bacterial genomes. To assess whether sequencing depth was sufficient for confident detection, we performed a rarefaction analysis in the revised manuscript using the breadth (fraction of genome covered by reads) and coverage (average read depth) of assembled MAGs. This approach directly evaluates genome detection completeness and avoids biases from marker-gene-based taxonomic profiling (e.g., MetaPhlAn). In this context, coverage serves as a proxy for sampling effort, while breadth serves as a measure of genome detection (<https://doi.org/10.1038/s41587-020-00797-0>). Plotting breadth against coverage allows us to evaluate whether increasing sequencing depth would yield additional genomic information. Plotting breadth against coverage revealed a knee at ~ 0.34 and a plateau near ~ 0.85 , indicating that additional sequencing would yield minimal new genomic information. This suggests that sequencing depth was sufficient to capture most detectable genomes, including low-abundance taxa with recommended thresholds of breadth > 0.5 and coverage > 5 (as discussed in <https://doi.org/10.1038/s41587-020-00797-0>). Furthermore, to address the reviewer's concern, we performed a sensitivity analysis to explore the impact of varying thresholds. Using the recommended thresholds (breadth > 0.5 and coverage > 5), our analysis recapitulates the read-based findings, indicating that *E. coli* is highly likely to be detected in infants at both two and six months postdelivery. To further evaluate the robustness of our criteria, we performed a sensitivity analysis across a range of thresholds (breadth = 0.1–1.0; coverage = 0–20). *E. coli* was consistently detected across samples up to the plateau, particularly at two and six months post-delivery. Detection only declined above ~ 0.85 breadth, which likely

reflects incomplete MAG assembly or uneven coverage rather than true absence. This analysis supports our approach, which balances sensitivity and specificity, capturing ecologically relevant populations while avoiding spurious detections. Similarly, our 0.1% relative abundance cutoff in the read-based analysis was chosen to exclude ecologically insignificant background signals. Although *E. coli* could theoretically occur below this threshold, such minimal abundances are unlikely to support cross-feeding or other functional interactions. The combination of read-based and genome-resolved evidence, together with the observed coverage-breadth plateau, strongly indicates that sequencing depth was adequate for robust detection of *E. coli* and its ecological associations with HMO-utilizing species.

Finally, while we agree with the reviewer that open questions remain regarding the in vivo relevance of the observed mutualism, we believe that additional qPCR screening for very rare *E. coli* would not substantially advance the ecological interpretation. Direct validation of metabolic exchange in vivo would instead require stable isotope-labeled HMO experiments to trace carbon flux across taxa. This is technically demanding and highly expensive, but a promising direction for future work. We have clarified these limitations and the rationale for our abundance thresholds in the revised results as described above, and in our revised discussion (revised manuscript lines: 438-444 and 477-480).

Reviewer response:

I do not agree that low abundant taxa (<0.1%) are ecologically negligible, nor that crossfeeding reactions are irrelevant just because you cross a certain threshold (of relative abundance). Archaea (PMID: 38292760) and fungal (PMID: 39676474) species are good examples of functionally important taxa that are not very abundant in the gut. Fungal species are typically only between 0.01-0.1%, but important in cross-kingdom interactions. No doubt that *Bifidobacterium* and *Escherichia* co-exist in the gut of most breastfed infants, but I do not think that the author's data fully supports their conclusions: "E. coli depends on monosaccharides released during extracellular hydrolysis by primary HMO-degraders" (l. 33-34)

"To gain simple sugars in the infant gut, E. coli relies entirely on the extracellular degradation of HMOs by primary degraders like B. bifidum, exploiting the breakdown products released without incurring the metabolic costs of substrate degradation. In formula-fed infants, readily available sugars like lactose allow E. coli to grow independently of primary HMO degraders, explaining its elevated abundances in these populations. In exclusively breastfed infants, the success of E. coli is dependent on the activity and abundance of the primary degrader, its capture efficiency of HMO-derived byproducts, and the metabolic costs associated with producing HMO-degrading enzymes." (l. 445-452).

Although the in vitro studies show that *E. coli* can indeed cross-feed on *B. bifidum* released HMO-remnants, I'm not convinced that this is in fact what is going on in breastfed infants:

1) If “*E. coli* relies entirely on the extracellular degradation of HMOs by primary degraders like *B. bifidum*”, then it wouldn’t be possible to detect it without *Bifidobacterium* species – but this is in fact observed in 5-20% of the infants at 2m and 6m (Fig. 1E).

2) As the authors now mention in the discussion, *E. coli* can grow on lactose in formula milk – this is also very likely to happen in breastfed infants, explaining *E. coli*’s high prevalence and co-occurrence with HMO-degrading *Bifidobacterium* species (in general – not just *B. bifidum*).

Author response:

We appreciate the reviewer’s point that rare taxa can be functionally important and agree that our data do not demonstrate that *E. coli* relies on cross-feeding of HMO breakdown products *in vivo*. We have revised the entire manuscript accordingly, emphasizing lactose as the primary substrate for *E. coli* in the infant gut and HMO breakdown products as a plausible secondary source of carbohydrates, given the presence of an extracellular HMO degrader such as *B. bifidum*.

Regarding the 0.1% relative abundance threshold, we now clarify in the revised manuscript that this cutoff was chosen not to imply ecological irrelevance of rare taxa, but to ensure statistical robustness (revised manuscript lines: 104-106). This helps us reduce false-positive artifacts in sparse and zero-inflated microbiome data. However, we explicitly acknowledge the absence of qPCR and its implications for interpreting very low-abundance Enterobacteriaceae (revised manuscript lines 470-474).

Furthermore, the authors group all “Bifidobacterium dominant” samples (Fig. 1E), without stratification into specific Bifidobacterium species, yet they only focus on *B. bifidum* for the later in vitro experiments (see point below).

2) Based on the human gut microbiome data, the authors suggest that HMO-utilizing Bifidobacterium species cross-feed HMO-degradants to *E. coli*. Yet, this is very unlikely to be true for other Bifidobacterium species than *B. bifidum*, since *B. longum* and *B. breve* (in contrast to *B. bifidum*) strains mostly employ intracellular HMO-degradation strategies (PMID: 31888048) and therefore will not liberate and cross-feed HMO sugars to *E. coli*. This is a problem and creates a disconnection between the human data and the in vitro data.

We thank the reviewer for this useful comment. In the revised manuscript, we clarify the rationale for focusing on Bifidobacterium bifidum as described below. Although *B. bifidum* is the best-characterized extracellular HMO degrader, recent studies indicate that certain *B. breve* and *B. longum* strains can also employ extracellular or semiextracellular degradation mechanisms (doi: 10.1093/femsre/fuad056 and <https://doi.org/10.1038/srep38560>). Thus, potential cross-feeding with *E. coli* may not be exclusive to *B. bifidum*. We have expanded the discussion in the revised manuscript to reflect this nuance and to highlight that our findings with *B. bifidum* represent one well-resolved example within a broader ecological context.

We focused our in vitro validation on *B. bifidum* because it displayed distinct ecological and genomic signatures in our dataset. Specifically, (i) significantly lower gene-level nucleotide diversity compared to *B. breve* and *B. longum* (Fig. 2C), and (ii) GH2-associated genomic traits consistent with extracellular specialization (Fig. 3A). Furthermore, *B. bifidum* was the only species that consistently grew in minimal medium using 2'FL as the sole carbon source. Establishing these conditions was technically challenging but a critical prerequisite for mechanistic testing.

Given these findings, potential cross-feeding interactions between *E. coli* and these bifidobacterial species may not be exclusive to *B. bifidum* and warrant further investigation. While exploring these interactions is beyond the scope of the current study, we acknowledge the importance of this avenue for future research. We have expanded the discussion in the revised manuscript to address this point and hope the reviewer finds our perspective reasonable. Specifically, we highlight now that our data demonstrates one clear example of mutualistic interaction between *B. bifidum* and *E. coli*, while leaving open the possibility that other Bifidobacterium species may participate in analogous interactions. (revised manuscript lines: 411-421)

Additionally, in response to the reviewer’s suggestion, we have now included a species level stratification of Bifidobacterium taxa exceeding 20% relative abundance per sample (revised Supplementary Fig. 1). This analysis reveals that *B. bifidum*, *B. longum* subsp. *longum*, and *B. breve* each contribute to the observed co-occurrence patterns with *E. coli*, albeit to varying degrees.

Reviewer response:

To my knowledge extracellular HMO degradation has not been proven in *B. breve*, and in *B. longum* it is confined only to LNT, with a low prevalence (PMID: 31888048);

PMID: 38206006). In my opinion it is unlikely that the HMO cross-feeding between *B. bifidum* and *E. coli* would also apply to the other HMO-degrading Bifidobacterium species, but the authors could of course prove me wrong.

The authors' new data clearly shows that co-occurrence of *E. coli* with Bifidobacterium species is not unique to *B. bifidum*, but also observed with *B. breve* and *B. longum*. This underlines my point of a disconnection between the observation for human data and the in vitro work and suggests that other mechanisms such as *E. coli* availability of lactose in the gut of breastfed infants are explaining the common co-occurrence between HMO-degrading Bifidobacterium species and *E. coli*.

Author response:

We thank the reviewer for their thoughtful assessment. In the revised manuscript we now reframe the ecological interpretation: lactose is presented as the primary substrate sustaining *E. coli* in breastfed infants, while cross-feeding of HMO-degradation products may additionally modulate *E. coli* abundances, given the presence of extracellular HMO-degraders. We agree with the reviewer that robust extracellular HMO degradation is best established for *B. bifidum*, while it remains speculative and poorly explored for other bifidobacteria. In the revised manuscript, we argue that utilization of both carbohydrate pools is not mutually exclusive and integrate this perspective into our discussion of how aspects of cooperation and competition shape the stability and resilience of the infant gut microbiota.

3) The authors claim (l. 37-39) that *E. coli* relies on primary HMO degraders to persist in the gut, yet they do not include any formula-fed infants as controls AND previous studies show that *E. coli* prevalence and abundance is in fact even higher in formula fed infants (PMID: 15668012).

We thank the reviewer for this comment and for citing the relevant work, which includes one of our co-authors. It is correct that *E. coli* prevalence and abundance can be higher in formula-fed infants. This difference likely arises because formula contains higher levels of free disaccharides, primarily lactose, which *E. coli* can utilize directly. In this context, *E. coli* and bifidobacteria compete for readily available sugars, and primary HMO degradation is not required. In contrast, in breastfed infants, *E. coli* relies on substrates liberated by HMO-degrading bifidobacteria, consistent with our findings. We have clarified this further in our discussion in the revised manuscript. Furthermore, we note that we deliberately focused exclusively on healthy, breastfed infants, as such cohorts are rare and particularly valuable for studying HMO-driven microbial interactions. We have extended the discussion to highlight this point in the revised manuscript lines 447-449.

Reviewer response:

Formula milk usually contains similar levels of lactose as in human milk. So, if this is the explanation for the common presence and abundance of *E. coli* in formula-fed infants, this should also apply for breastfed infants.

Author response:

We have revised our discussion on this topic (revised manuscript lines 409-424). Briefly, HMOs uniquely shape the breastfed gut by favoring *Bifidobacterium*. In formula-fed infants, the absence of HMOs removes this advantage and shifts metabolism toward simple sugars, which *E. coli* metabolizes faster.

Minor comments

L. 124-133. I don't understand this way of using Shannon index. Shannon index tells you something about species diversity in a sample – it doesn't make sense to do for individual species? Are you looking at Shannon index of all species in samples where the indicated species are present?

We thank the reviewer for pointing this out. We agree that the description in the original text was unclear. We did not compute a Shannon index per species within a sample. What we did was to calculate the Shannon diversity of each species' abundance distribution across all samples. This measure reflects how evenly, or consistently a species occurs across individuals in cohort. And we did this to figure out which species are the most consistently occurring. We have revised this in the manuscript accordingly. (revised manuscript lines 120-128).

Reviewer response:

Ok. This should be described in the methods.

Author response:

We have now added the relevant methodological information in revised manuscript lines: 636-637.

L. 134-140. How can you rule out that all samples in fact have E. coli present (but just at rel abundance below <0.1%)? Is the statistical analysis based on both 2m and 6m data? What happen if you include only 2m or 6m? What about 11m?

We thank the reviewer for pointing this out. Double checking this has improved our manuscript. In fact, we did not separate this analysis by age groups, which was an oversight. We now calculate a Fisher's exact test only using samples from infants B-02, B-06, and B-11 (previously maternal ones were included). Also, we have conducted a sensitivity analysis showing that within any threshold filter between 0-1% relative abundance, this relationship between E. coli and bifidobacteria remains significant, therefore stable. (revised manuscript lines 131-136)

Reviewer response:

Ok. But what about stratified analyses, including only 2m, 6m OR 11m data?

Author response:

We appreciate the request for stratified analyses by age (2m, 6m, 11m). After re-running Fisher's exact tests within each stratum, we found that the data structure makes formal hypothesis testing unfeasible. Before weaning (2m and 6m), E. coli (>0.1% rel. abundance) and Bifidobacterium dominance (>20%) are almost universally present, creating zero or near-zero cells in the contingency tables. This yields inconclusive odds ratios with extremely wide confidence intervals. Pooling ages would inappropriately mix biologically distinct periods, while stratification suffers from low power. Therefore, we now provide a visual representation of the age-stratified contingency tables (revised Fig. S1E), which shows that, pre-weaning, samples with highly abundant Bifidobacterium almost always also have detectable E. coli. We hope this illustration clarifies that, pre-weaning, highly abundant Bifidobacterium and E. coli (>0.1%) co-occur in the vast majority of samples.

L. 138-143. I'm not sure I agree that 20% rel abundance equals dominance. In my view this would mean >50%. I assume that using this definition of dominance (>20%), implies that no other bacterial species in the sample reached higher rel abundance? Please elaborate.

We agree with the reviewer's concern here and will rephrase dominance to 'highly abundant (> 20%)'

Reviewer response:

Ok.

Fig. 1. Panel B. Chao-1 estimates richness, so I guess it is not "observed diversity"? Did you look at actual Observed species (at a certain seq depth cutoff) and/or Shannon index? What about evenness? Chao estimated richness seems quite low, especially considering maternal samples with only 20-40 species?

We thank the reviewer for this comment. The y-axis label in the original version of Fig. 1B was incorrect. We have now corrected this panel and replaced it with Shannon diversity, capturing both richness and evenness. Additionally, we have added a rankabundance curve in the revised manuscript to better visualize the observed diversity in our sample set of infants and mothers. (revised manuscript lines 102-104 and revised Figure S1A)

Reviewer response:

Ok.

Major concerns

1) The authors claim that *E. coli* presence in the infant gut depends on having a major dominating HMO-utilizing species (in most cases a Bifidobacterium species). However, *E. coli* is very prevalent in the cohort (76-96%) and the authors use a cutoff for presence/absence of 0.1% relative abundance. It is not unlikely that *E. coli* is in fact present in all samples but either below the selected cutoff or just right below the limit of detection of sequencing. Without more sensitive methods to determine presence/absence (e.g. qPCR using *E. coli* specific primers), I am not confident that the author's conclusion holds true.

We thank the reviewer for this thoughtful comment. While it is true that *E. coli* might be present in samples below our detection threshold, we consider that presence at such low abundances (<0.1%) would ecologically be negligible. Cross-feeding interactions require sufficient biomass to generate metabolite concentrations that would sustain growth of both cross-feeding partners. Thus, populations below this level are unlikely to participate meaningfully in community dynamics. We have clarified this in the revised manuscript.

We appreciate the reviewer's concern, and we conducted additional analyses in the revised manuscript in response, aiming to underscore the validity of our approach [revised manuscript lines 220-228 and revised Figure S3]. Unfortunately, we no longer have access to the original samples for qPCR quantification (as remaining samples were destroyed due to ethical/MTA requirements), and the limited archived DNA we hold, stored at -20°C , is unlikely to yield reliable results. However, we emphasize that shotgun metagenomics provides a highly sensitive and quantitative approach for detecting bacterial genomes. To assess whether sequencing depth was sufficient for confident detection, we performed a rarefaction analysis in the revised manuscript using the breadth (fraction of genome covered by reads) and coverage (average read depth) of assembled MAGs. This approach directly evaluates genome detection completeness and avoids biases from marker-gene-based taxonomic profiling (e.g., MetaPhlAn). In this context, coverage serves as a proxy for sampling effort, while breadth serves as a measure of genome detection (<https://doi.org/10.1038/s41587-020-00797-0>). Plotting breadth against coverage allows us to evaluate whether increasing sequencing depth would yield additional genomic information. Plotting breadth against coverage revealed a knee at ~ 0.34 and a plateau near ~ 0.85 , indicating that additional sequencing would yield minimal new genomic information. This suggests that sequencing depth was sufficient to capture most detectable genomes, including low-abundance taxa with recommended thresholds of breadth > 0.5 and coverage > 5 (as discussed in <https://doi.org/10.1038/s41587-020-00797-0>). Furthermore, to address the reviewer's concern, we performed a sensitivity analysis to explore the impact of varying thresholds. Using the recommended thresholds (breadth > 0.5 and coverage > 5), our analysis recapitulates the read-based findings, indicating that *E. coli* is highly likely to be detected in infants at both two and six months postdelivery. To further evaluate the robustness of our criteria, we performed a sensitivity analysis across a range of thresholds (breadth = 0.1–1.0; coverage = 0–20). *E. coli* was consistently detected across samples up to the plateau, particularly at two and six months post-delivery. Detection only declined above ~ 0.85 breadth, which likely

reflects incomplete MAG assembly or uneven coverage rather than true absence. This analysis supports our approach, which balances sensitivity and specificity, capturing ecologically relevant populations while avoiding spurious detections.

Similarly, our 0.1% relative abundance cutoff in the read-based analysis was chosen to exclude ecologically insignificant background signals. Although *E. coli* could theoretically occur below this threshold, such minimal abundances are unlikely to support cross-feeding or other functional interactions. The combination of read-based and genome-resolved evidence, together with the observed coverage-breadth plateau, strongly indicates that sequencing depth was adequate for robust detection of *E. coli* and its ecological associations with HMO-utilizing species.

Finally, while we agree with the reviewer that open questions remain regarding the *in vivo* relevance of the observed mutualism, we believe that additional qPCR screening for very rare *E. coli* would not substantially advance the ecological interpretation. Direct validation of metabolic exchange *in vivo* would instead require stable isotope-labeled HMO experiments to trace carbon flux across taxa. This is technically demanding and highly expensive, but a promising direction for future work. We have clarified these limitations and the rationale for our abundance thresholds in the revised results as described above, and in our revised discussion (revised manuscript lines: 438-444 and 477-480).

Reviewer response:

I do not agree that low abundant taxa (<0.1%) are ecologically negligible, nor that cross-feeding reactions are irrelevant just because you cross a certain threshold (of relative abundance). Archaea (PMID: 38292760) and fungal (PMID: 39676474) species are good examples of functionally important taxa that are not very abundant in the gut. Fungal species are typically only between 0.01-0.1%, but important in cross-kingdom interactions.

No doubt that Bifidobacterium and Escherichia co-exist in the gut of most breastfed infants, but I do not think that the author's data fully supports their conclusions:

"E. coli depends on monosaccharides released during extracellular hydrolysis by primary HMO-degraders" (l. 33-34)

"To gain simple sugars in the infant gut, E. coli relies entirely on the extracellular degradation of HMOs by primary degraders like B. bifidum, exploiting the breakdown products released without incurring the metabolic costs of substrate degradation. In formula-fed infants, readily available sugars like lactose allow E. coli to grow independently of primary HMO degraders, explaining its elevated abundances in these populations. In exclusively breastfed infants, the success of E. coli is dependent on the activity and abundance of the primary degrader, its capture efficiency of HMO-derived byproducts, and the metabolic costs associated with producing HMO-degrading enzymes." (l. 445-452)

Although the *in vitro* studies show that *E. coli* can indeed cross-feed on *B. bifidum* released HMO-remnants, I'm not convinced that this is in fact what is going on in breastfed infants:

- 1) If “*E. coli* relies entirely on the extracellular degradation of HMOs by primary degraders like *B. bifidum*”, then it wouldn’t be possible to detect it without Bifidobacterium species – but this is in fact observed in 5-20% of the infants at 2m and 6m (Fig. 1E).
- 2) As the authors now mention in the discussion, *E. coli* can grow on lactose in formula milk – this is also very likely to happen in breastfed infants, explaining *E. coli*’s high prevalence and co-occurrence with HMO-degrading Bifidobacterium species (in general – not just *B. bifidum*).

Furthermore, the authors group all “Bifidobacterium dominant” samples (Fig. 1E), without stratification into specific Bifidobacterium species, yet they only focus on *B. bifidum* for the later in vitro experiments (see point below).

2) Based on the human gut microbiome data, the authors suggest that HMO-utilizing Bifidobacterium species cross-feed HMO-degradants to *E. coli*. Yet, this is very unlikely to be true for other Bifidobacterium species than *B. bifidum*, since *B. longum* and *B. breve* (in contrast to *B. bifidum*) strains mostly employ intracellular HMO-degradation strategies (PMID: 31888048) and therefore will not liberate and cross-feed HMO sugars to *E. coli*. This is a problem and creates a disconnection between the human data and the in vitro data.

We thank the reviewer for this useful comment. In the revised manuscript, we clarify the rationale for focusing on *Bifidobacterium bifidum* as described below. Although *B. bifidum* is the best-characterized extracellular HMO degrader, recent studies indicate that certain *B. breve* and *B. longum* strains can also employ extracellular or semi-extracellular degradation mechanisms (doi: 10.1093/femsre/fuad056 and <https://doi.org/10.1038/srep38560>). Thus, potential cross-feeding with *E. coli* may not be exclusive to *B. bifidum*. We have expanded the discussion in the revised manuscript to reflect this nuance and to highlight that our findings with *B. bifidum* represent one well-resolved example within a broader ecological context.

We focused our *in vitro* validation on *B. bifidum* because it displayed distinct ecological and genomic signatures in our dataset. Specifically, (i) significantly lower gene-level nucleotide diversity compared to *B. breve* and *B. longum* (Fig. 2C), and (ii) GH2-associated genomic traits consistent with extracellular specialization (Fig. 3A). Furthermore, *B. bifidum* was the only species that consistently grew in minimal medium using 2’FL as the sole carbon source. Establishing these conditions was technically challenging but a critical prerequisite for mechanistic testing.

Given these findings, potential cross-feeding interactions between *E. coli* and these bifidobacterial species may not be exclusive to *B. bifidum* and warrant further investigation. While exploring these interactions is beyond the scope of the current study, we acknowledge the importance of this avenue for future research. We have expanded the discussion in the revised manuscript to address this point and hope the reviewer finds our perspective reasonable. Specifically, we highlight now that our data demonstrates one clear example of mutualistic interaction between *B. bifidum* and *E. coli*, while leaving open the possibility that other *Bifidobacterium* species may participate in analogous interactions. (revised manuscript lines: 411-421)

Additionally, in response to the reviewer’s suggestion, we have now included a species level stratification of *Bifidobacterium* taxa exceeding 20% relative abundance per sample (revised Supplementary Fig. 1). This analysis reveals that *B. bifidum*, *B. longum*

subsp. longum, and *B. breve* each contribute to the observed co-occurrence patterns with *E. coli*, albeit to varying degrees.

Reviewer response:

To my knowledge extracellular HMO degradation has not been proven in *B. breve*, and in *B. longum* it is confined only to LNT, with a low prevalence (PMID: 31888048; PMID: 38206006). In my opinion it is unlikely that the HMO cross-feeding between *B. bifidum* and *E. coli* would also apply to the other HMO-degrading Bifidobacterium species, but the authors could of course prove me wrong.

The authors' new data clearly shows that co-occurrence of *E. coli* with Bifidobacterium species is not unique to *B. bifidum*, but also observed with *B. breve* and *B. longum*. This underlines my point of a disconnection between the observation for human data and the in vitro work and suggests that other mechanisms such as *E. coli* availability of lactose in the gut of breastfed infants are explaining the common co-occurrence between HMO-degrading Bifidobacterium species and *E. coli*.

3) The authors claim (l. 37-39) that *E. coli* relies on primary HMO degraders to persist in the gut, yet they do not include any formula-fed infants as controls AND previous studies show that *E. coli* prevalence and abundance is in fact even higher in formula fed infants (PMID: 15668012).

We thank the reviewer for this comment and for citing the relevant work, which includes one of our co-authors. It is correct that *E. coli* prevalence and abundance can be higher in formula-fed infants. This difference likely arises because formula contains higher levels of free disaccharides, primarily lactose, which *E. coli* can utilize directly. In this context, *E. coli* and bifidobacteria compete for readily available sugars, and primary HMO degradation is not required. In contrast, in breastfed infants, *E. coli* relies on substrates liberated by HMO-degrading bifidobacteria, consistent with our findings. We have clarified this further in our discussion in the revised manuscript. Furthermore, we note that we deliberately focused exclusively on healthy, breastfed infants, as such cohorts are rare and particularly valuable for studying HMO-driven microbial interactions. We have extended the discussion to highlight this point in the revised manuscript lines 447-449.

Reviewer response:

Formula milk usually contains similar levels of lactose as in human milk. So, if this is the explanation for the common presence and abundance of *E. coli* in formula-fed infants, this should also apply for breastfed infants.

Minor comments

L. 124-133. I don't understand this way of using Shannon index. Shannon index tells you something about species diversity in a sample – it doesn't make sense to do for individual species? Are you looking at Shannon index of all species in samples where the indicated species are present?

We thank the reviewer for pointing this out. We agree that the description in the original

text was unclear. We did not compute a Shannon index per species within a sample. What we did was to calculate the Shannon diversity of each species' abundance distribution across all samples. This measure reflects how evenly, or consistently a species occurs across individuals in cohort. And we did this to figure out which species are the most consistently occurring. We have revised this in the manuscript accordingly. (revised manuscript lines 120-128).

Reviewer response:

Ok. This should be described in the methods.

L. 134-140. How can you rule out that all samples in fact have *E. coli* present (but just at rel abundance below $<0.1\%$)? Is the statistical analysis based on both 2m and 6m data? What happens if you include only 2m or 6m? What about 11m?

We thank the reviewer for pointing this out. Double checking this has improved our manuscript. In fact, we did not separate this analysis by age groups, which was an oversight. We now calculate a Fisher's exact test only using samples from infants B-02, B-06, and B-11 (previously maternal ones were included). Also, we have conducted a sensitivity analysis showing that within any threshold filter between 0-1% relative abundance, this relationship between *E. coli* and bifidobacteria remains significant, therefore stable. (revised manuscript lines 131-136)

Reviewer response:

Ok. But what about stratified analyses, including only 2m, 6m OR 11m data?

L. 138-143. I'm not sure I agree that 20% rel abundance equals dominance. In my view this would mean $>50\%$. I assume that using this definition of dominance ($>20\%$), implies that no other bacterial species in the sample reached higher rel abundance? Please elaborate.

We agree with the reviewer's concern here and will rephrase dominance to 'highly abundant ($> 20\%$)'

Reviewer response:

Ok.

Fig. 1. Panel B. Chao-1 estimates richness, so I guess it is not "observed diversity"? Did you look at actual Observed species (at a certain seq depth cutoff) and/or Shannon index? What about evenness? Chao estimated richness seems quite low, especially considering maternal samples with only 20-40 species?

We thank the reviewer for this comment. The y-axis label in the original version of Fig. 1B was incorrect. We have now corrected this panel and replaced it with Shannon diversity, capturing both richness and evenness. Additionally, we have added a rankabundance curve in the revised manuscript to better visualize the observed diversity in our sample set of infants and mothers. (revised manuscript lines 102-104 and revised Figure S1A)

Reviewer response:

Ok.